# RESTORATION BASED GENERATIVE MODELS

## ABSTRACT

Denoising diffusion models (DDMs) have recently attracted increasing attention by showing impressive synthesis quality. DDMs are built on a diffusion process that pushes data to the noise distribution and the models learn to denoise. In this paper, we establish the interpretation of DDMs in terms of image restoration (IR). Integrating IR literature allows us to use an alternative objective and diverse forward processes, not confining to the diffusion process. By imposing prior knowledge on the loss function grounded on MAP estimation, we eliminate the need for the expensive sampling of DDMs. Also, we propose a multi-scale training, which improves the performance compared to the diffusion process, by taking advantage of the flexibility of the forward process. Our model improves the quality and efficiency of both training and inference, Furthermore, we show the applicability of our model to inverse problems. We believe that our framework paves the way for designing a new type of flexible general generative model.

## 1 INTRODUCTION

Generative modeling is a prolific machine learning task that the models learn to describe how a dataset is distributed and generate new samples from the distribution. The most widely used generative models primarily differ in their choice of bridging the data distribution to a tractable latent distribution (Goodfellow et al., 2020; Kingma & Welling, 2014; Rezende et al., 2014; Rezende & Mohamed, 2015; Sohl-Dickstein et al., 2015; Chen et al., 2021a). In recent years, denoising diffusion models (DDMs) (Ho et al., 2020; Song & Ermon, 2019; Song et al., 2020b; Dockhorn et al., 2021) have drawn considerable attention by demonstrating remarkable results in terms of both high sample quality and likelihood. DDMs rely on a forward diffusion process that progressively transforms the data into Gaussian noise, and they learn to reverse the noising process. Albeit their enormous successes, their forward process is fixed as a diffusion process, which gives rise to a few limitations. To

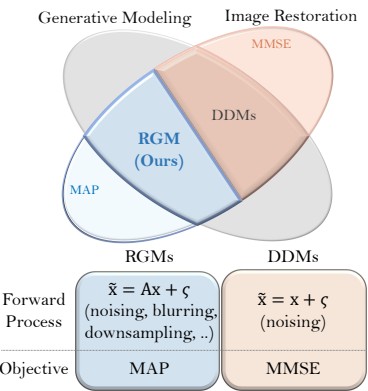

Figure 1: Comparison of DDMs and RGMs.

pull latent variables back to the data distribution, the denoising process requires thousands of network evaluations to sample a single instance. Many follow-up studies consider enhancing inference speed (Song et al., 2020a; Jolicoeur-Martineau et al., 2021; Tachibana et al., 2021) or grafting with other generative models (Xiao et al., 2021a; Vahdat et al., 2021; Zhang & Chen, 2021; Pandey et al., 2022).

In this study, we focus on a different perspective. We interpret the DDMs through the lens of image restoration (IR), which is a family of inverse problems for recovering the original images from corrupted ones (Castleman, 1996; Gunturk & Li, 2018). The corruption arises in various forms, including noising (Buades et al., 2005; Rudin et al., 1992), blurring (Biemond et al., 1990), and downsampling (Farsiu et al., 2004). IR has been a long-standing problem because of its high practical value in various applications (Besag et al., 1991; Banham & Katsaggelos, 1997; Lehtinen et al., 2018; Ma et al., 2011). From an IR point of view, DDMs can be considered as IR models based on minimum mean square error (MMSE) estimation (Zervakis & Venetsanopoulos, 1991; Laumont et al., 2022), focusing only on the denoising task. Mathematically, IR is an ill-posed inverse problem in the sense that it does not admit a unique solution and hence, leads to instability in reconstruction (Hadamard, 1902). Owing to the ill-posedness of IR, MMSE which only measures data fidelity

produces impertinent results. DDMs alleviate this problem by leveraging costly Langevin sampling, and this inefficient inference scheme has been regarded as an indispensable tool in the literature of DDMs. By casting DRMs as IR models, however, the forward process need not be restricted to Gaussian noising, and ill-posedness can be detoured in ways other than Langevin dynamics.

Inspired by this observation, we propose a new flexible family of generative models, that we refer to as *restoration-based generative models (RGMs)*. First, we adopt an alternative objective; a maximum a posteriori (MAP) (Trussell, 1980; Hunt, 1977), which is predominantly used in IR. The MAP-based estimator compensates the ill-posedness by regularizing the data fidelity loss by a prior term. Many advent hand-crafted regularization schemes (Tikhonov, 1963; Donoho, 1995; Mallat, 1999; Baraniuk, 2007) encourage solutions to satisfy certain properties, such as smoothness and sparsity. However, for the purpose of density estimation, we implicitly parameterize a prior term as a variational regularization via GAN (Goodfellow et al., 2020) with a newly introduced random auxiliary variable. Our MAP approach retains the density estimating capability of DDMs at a much smaller computation cost. Secondly, unlike DDMs, which are buried in a Gaussian noising process, RGMs can be combined with other general degradation processes. As one instantiation, we design a multi-scale training that resolves the latent inefficiency of DDMs. Because the behavior of generative models is significantly affected by how the data distribution is transformed into a simple distribution, our approach opens the way for designing more flexible generative models. Our comprehensive empirical studies on image generation and inverse problems demonstrate that RGMs generate samples rivaling the quality of DDMs. Also, the inference of our model is several orders of magnitude faster than DDMs. In particular, our model achieve FID 2.47 on CIFAR10, with only seven number of network function evaluations.

## 2 BACKGROUND

**Image Restoration**    A common inverse problem arising in image processing, including denoising, deblurring, super-resolution, and inpainting, is the estimation of an image $\mathbf{x}$ given a corrupted image

$$\mathbf{y} = \mathbf{A}\mathbf{x} + \xi, \tag{1}$$

where $\mathbf{A}$ is a matrix that models the degradation process, including blurring and downsampling kernels, and $\xi \sim \mathcal{N}(0, \mathbf{\Sigma})$ is an additive noise. A family of such problems are known as image restoration (IR). The inference of the image $\mathbf{x}$ from the noised one $\mathbf{y}$ is typically ill-posed, in the sense that the inverse problem (1) has multiple valid explanations (Hadamard, 1902). In other words, the noisy $\mathbf{y}$ does not have exactly one restoration $\mathbf{x}$. This is further exacerbated when the noise level is large. To produce consistent results, most methods use the maximum a posteriori (MAP) estimator:

$$\mathbf{x}_{\text{MAP}}^* = \operatorname{argmax}_{\mathbf{x}} \, \log p(\mathbf{x} \mid \mathbf{y}) = \operatorname{argmin}_{\mathbf{x}} f(\mathbf{x}, \mathbf{y}) + \lambda g(\mathbf{x}), \tag{2}$$

where $f(\mathbf{x}, \mathbf{y}) = -\log p(\mathbf{y} \mid \mathbf{x}) = \frac{1}{2}\left\|(\mathbf{\Sigma}^\dagger)^{\frac{1}{2}}(\mathbf{A}\mathbf{x} - \mathbf{y})\right\|_2^2$ is the data fidelity term with the pseudoinverse (Moore, 1920) $\mathbf{\Sigma}^\dagger$, $g$ is the prior term that encourages the reconstruction to satisfy some prior assumptions on $\mathbf{x}$, and a scalar $\lambda \geq 0$ controls the strength of the regularization. The regularization term $g$ is essential because it relieves the ill-posedness nature of the inverse problem by imposing the assumption about the desirable solution. Therefore, many researchers have been devoted to designing a proper $g$ (Rudin et al., 1992; Mallat, 1999; Lunz et al., 2018).

**Denoising Generative Models**    Denoising diffusion models (DDMs), such as DDPM (Ho et al., 2020), and score matching with Langevin dynamics (Song et al., 2020b), have recently emerged as the forefront of image synthesis research. Starting from the data distribution, they gradually corrupt the image $\mathbf{x}_0 \sim p_{\text{data}}$ into Gaussian noise over time through a forward Markovian diffusion process;

$$q(\mathbf{x}_{1:T} \mid \mathbf{x}_0) = \prod_{t=0}^{T-1} q^{(t)}(\mathbf{x}_{t+1} \mid \mathbf{x}_t), \ \mathbf{x}_0 \sim p_{\text{data}}. \tag{3}$$

They pose the data generation as an iterative denoising procedure $p_\theta^{(t)}(\mathbf{x}_t \mid \mathbf{x}_{t+1})$, the reverse of the forward diffusion process:

$$p_\theta(\mathbf{x}_{0:T}) = p^{(T)}(\mathbf{x}_T) \prod_{t=0}^{T-1} p_\theta^{(t)}(\mathbf{x}_t \mid \mathbf{x}_{t+1}), \ \mathbf{x}_T \sim \mathcal{N}(\mathbf{0}, \mathbf{I}). \tag{4}$$

As they use linear diffusion process whose diffusion distributions $q^{(t)}(\mathbf{x}_{t+1} \mid \mathbf{x}_t)$ are modeled with conditional Gaussian distributions, we can deduces a tractable evidence lower bound (ELBO) (Sohl-Dickstein et al., 2015). The ELBO can be further simplified to the following objective (Ho et al., 2020; Song et al., 2020b):

$$\mathcal{L}(\theta) = \Sigma_{t=0}^{T} \mathbb{E}_{\mathbf{x}_0 \sim p_{\text{data}}, \mathbf{x}_t \sim q_{\sigma_t}(\mathbf{x}_t \mid \mathbf{x}_0)} \left[ \lambda(t) \| G_\theta(\mathbf{x}_t, t) - \mathbf{x}_0 \|_2^2 \right], \tag{5}$$

where $G_\theta$ is a neural network parametrized by $\theta$ that learns the noise by minimizing (5), and $\lambda(t) \geq 0$.

## 3 METHOD

### 3.1 DDMS ARE RESTORATION MODELS

We open this section by drawing the interpretation of DDMs in terms of restoration. DDMs use VPSDE or VESDE (Song et al., 2020b) as a forward process, and these two are known to be exchangeable with each other (Kim et al., 2022). Therefore, the rest of the paper focuses on VESDE. For a given noise level $\sigma_t$, the forward process of the VESDE is formulated as

$$\mathbf{x}_t = \mathbf{x}_0 + \xi, \ \xi \sim \mathcal{N}(\mathbf{0}, \sigma_t^2 \mathbf{I}),$$

which is the forward process (1) with identity degradation matrix $\mathbf{A} = \mathbf{I}$. As $\nabla_{\mathbf{x}_t} \log q_{\sigma_t}(\mathbf{x}_t \mid \mathbf{x}_0) = -(\mathbf{x}_t - \mathbf{x}_0)/\sigma_t^2$, the loss (5) for each forward step can be rewritten as the following minimum mean square error (MMSE) objective:

$$\mathcal{L}(\theta) = \mathbb{E}_{\mathbf{x} \sim p_{\text{data}}, \mathbf{x}_t \sim \mathcal{N}(\mathbf{x}, \sigma_t^2 \mathbf{I})} \left[ \| G_\theta(\mathbf{x}_t, t) - \mathbf{x} \|_2^2 \right]. \tag{6}$$

Therefore, DDMs are IR models which seek a denoiser $G_\theta$ for each $\sigma_t$ that minimizes the MMSE (6). MMSE loss is simple and straightforward to train, however, it confronts some apparent drawbacks. Since the MMSE only contains the recovery term, the solution is only optimized to ensure accordance with the degradation process. Therefore, it is affected by the ill-posedness. To be precise, when $\sigma_t$ is large, (1) becomes a highly ill-posed and possess many solutions for a given observation. In this case, MMSE solution averages all these candidate solutions, resulting in an atypical reconstruction. Formally, the solution to the MMSE for a corrupted data $\mathbf{x}_t$ is

$$\mathbf{x}_{\text{MMSE}}^* = \int \mathbf{x} p_{\text{data}}(\mathbf{x}) p_{\sigma_t}(\mathbf{x}_t \mid \mathbf{x}) d\mathbf{x}.$$

Recent works (Laumont et al., 2022; Kawar et al., 2021) have endeavored to resolve this problem by stochastic sampling, however, they suffer from notoriously low efficiency as they roll out thousands of trajectories. In a similar manner, DDMs utilize a sampling scheme that requires thousands of steps. In summary, there are two limitations of DDMs from the IR perspective:

1. The degradation process is restricted to Gaussian noising.

2. The inference efficiency is intrinsically low due to the MMSE estimator.

In the following sections, we show how to cope with these two limitations.

### 3.2 MAP-BASED ESTIMATION FOR GENERATION

As alluded in Section 3.1, DDMs can be regarded as MMSE grounded IR models, specialized in denoising. This observation brings us a new perspective on the design of a family of flexible generative models. As an alternative to MMSE, we propose a new generative model based on the MAP (2):

$$\mathbb{E}_{\mathbf{x} \sim p_{\text{data}}, \mathbf{y} \sim \mathcal{N}(\mathbf{x}, \sigma^2 \mathbf{I})} \left[ \frac{1}{2\sigma^2} \| G_\theta(\mathbf{y}) - \mathbf{y} \|_2^2 + \lambda g(G_\theta(\mathbf{y}))) \right], \tag{7}$$

where the second term delivers the prior knowledge of data distribution. MAP has been adopted as a standard approach for high-dimensional imaging problems and is known to be more relevant than MMSE in many applications (Saha et al., 2009; Bigdeli et al., 2019; Chen, 2016). By leveraging prior information on the solution, MAP-based approaches alleviate the ill-posedness of the inverse problem (1), without use of costly sampling of MMSE estimation. Therefore, carefully crafting the relevant prior term is crucial. We now show how one can execute an appropriate prior term for density estimation while alleviating the ill-posedness.

**Alleviation of ill-posedness** Unlike the general denoising task, it is necessary to bridge the image to the Gaussian noise to learn the data distribution. As the noise level increases, a single distorted observation has several solutions, which indicates that the ill-posedness deepens. This in turn degrades the expressiveness of the model for estimating the distribution. Therefore, it is difficult for the regularization term to remedy all ill-posedness on its own. We further offload the ill-posedness by imposing a priori knowledge by introducing a random auxiliary variable $\mathbf{z} \sim \mathcal{N}(\mathbf{z} \mid \mathbf{0}, \mathbf{I})$. Importantly, $\mathbf{z}$ enables us to obtain several solutions for a heavily degraded $\mathbf{x}_t$ through the guidance provided by $\mathbf{z}$. This eventually helps to pull the noised observations back to the data distribution, allowing for a rich representation for the density.

**Implicit Prior Knowledge** For density estimation, the knowledge about the data distribution should be properly encoded in the prior term $g$ of (7). The explicit density function with determined marginal distribution is intractable, but it can be learned with the aid of well-developed generative models. That is, we can design $g$ by integrating various existing generative models. In this paper, we learn an implicit representation of the data density by adopting generative adversarial network (GAN), which has shown promising results in many generative tasks, and use generator loss as a relevant prior term $g$. For each forward step, our MAP-based objective for our generator in conjunction with the GAN prior is given by:

$$\mathcal{L}(G_\theta) = \mathbb{E}_{\mathbf{x} \sim p_{\text{data}}, \mathbf{y} \sim \mathcal{N}(\mathbf{x}, \sigma^2 \mathbf{I}), \mathbf{z} \sim \mathcal{N}(0, I)} \left[ \frac{1}{2\sigma^2} \|G_\theta(\mathbf{y}, \mathbf{z}) - \mathbf{y}\|_2^2 + \lambda g_\phi(G_\theta(\mathbf{y}, \mathbf{z})) \right], \quad (8)$$

where the first term is the data fidelity term, $g_\phi(\mathbf{x}) = \log(1 - D_\phi(\mathbf{x})) - \log D_\phi(\mathbf{x})$ is a learnable prior term that is trained coupled with a discriminator $D_\phi$, and $\lambda \geq 0$ is a hyperparameter. As conventional, we train the discriminator $D_\phi$ to minimize Jensen-Shannon divergence. Contrary to conventional MAP whose prior term is pre-defined, our approach tries to learn the prior term by coordinating with the denoiser $G_\theta$ through the discriminator. This end-to-end training allows our discriminative learning method to deliver promising performance. Note that when the discriminator is optimal, i.e. $D_\phi(x) = \frac{p(x)}{p(x) + q_\theta(x)}$, the loss (8) with $\lambda = 1$ agrees with

$$\mathbb{E}_{\mathbf{x} \sim p_{\text{data}}, \mathbf{y} \sim \mathcal{N}(\mathbf{x}, \sigma^2 \mathbf{I})} \left[ D_{KL}(q_\theta(\mathbf{x}|\mathbf{y}) \| p(\mathbf{x}|\mathbf{y})) \right] + \mathcal{H}(q_\theta), \quad (9)$$

where $\mathcal{H}$ denotes an entropy, which corresponds to training the model to learn the posterior distribution. The overall training procedure combined with all $\sigma \in \{\sigma_k\}_{k=1}^T$ is provided in Appendix B.2.

**Small Denoising Steps** A major downside of DDMs is their slow sampling procedure, which requires hundreds to thousands of denoising steps to obtain a single image. By adopting MAP approach and parametrizing the prior distribution through GAN, our model provides an avenue to offload the time-consuming sampling scheme and enables significantly small denoising steps. For small degradation we can obtain a restored image in one shot. But, as our restoration starts from the Gaussian noise, the data distribution is not completely estimated. Therefore, we perform the generation iteratively. In our experiments on CIFAR10, we generate a high-quality sample in four denoising steps, whereas most DDMs use hundreds to thousands steps.

### 3.3 EXTENSION TO GENERAL RESTORATION

In Section 3.2, we proposed a denoising generative model based on MAP-like objective. However, from the IR perspective, it is not necessary to restrict it to denoising ($\mathbf{A} = \mathbf{I}$) and it can be generalized to any family of degradation matrices $\mathbf{A}$ and noise factors $\boldsymbol{\Sigma}$ in (1). Utilizing the general forward process, we can learn the generative model by generalizing the loss function (8) as follows:

$$\mathcal{L}(G_\theta) = \mathbb{E}_{\mathbf{x} \sim p_{\text{data}}, \mathbf{y} \sim \mathcal{N}(\mathbf{A}\mathbf{x}, \boldsymbol{\Sigma}), \mathbf{z} \sim \mathcal{N}(\mathbf{0}, \mathbf{I})} \left[ \frac{1}{2} \left\| (\boldsymbol{\Sigma}^\dagger)^{\frac{1}{2}} (\mathbf{A} \cdot G_\theta(\mathbf{y}, \mathbf{z}) - \mathbf{y}) \right\|_2^2 + \lambda g_\phi(G_\theta(\mathbf{y}, \mathbf{z})) \right].$$

$$(10)$$

Therefore, RGM has an flexible structure that can permeate any forward process, and aids in designing a new generative model. Here, we propose a new model established upon super-resolution (SR).

**Multi-scale RGM** Most DDMs maintain the image size during the diffusion process by adding noise to individual pixels. Consequently, they are very inefficient because they require a latent as much as dimension of pixel space that is much larger than the submanifold of the image space. Motivated by

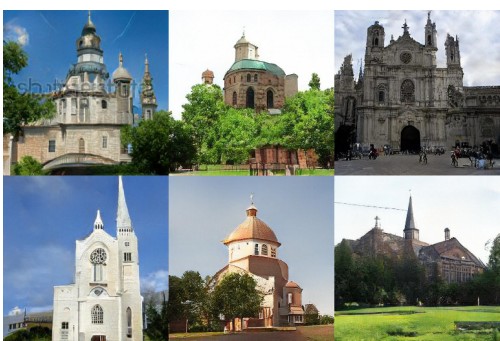 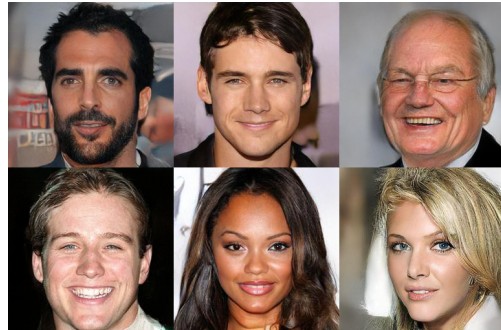

Figure 2: Generated samples on LSUN Church (left) and CelebA-HQ (right).

this, we take **A** as a block averaging filter that averages out $2 \times 2$ pixel values. Halving the image size at each coarsening step allows us a more expressive generative model with a lower-dimensional latent distribution. Moreover, multi-scale training has proven to be an effective strategy for synthesizing large scale images (Denton et al., 2015; Karras et al., 2017b; Reed et al., 2017). Therefore, our model produces strikingly realistic images by progressively extracting spatial information.

## 4 EXPERIMENTS

This section evaluates the performance of the proposed RGMs on several benchmark datasets, including CIFAR10 (Krizhevsky et al., 2009) ($32 \times 32$ unconditional), CelebA-HQ (Liu et al., 2015) ($256 \times 256$), and LSUN Church (Yu et al., 2015) ($256 \times 256$). We also show the capability of RGMs for solving inverse problems. We parametrize our generator $G_\theta$ based on the UNet-like structure (Ronneberger et al., 2015) which was successfully used in NCSN++ architecture (Xiao et al., 2021a). The internal details of the implementation can be found in Appendix B.

**Setup** We implement two models: *RGM-D* is a model trained with the diffusion process, which is mainly used by DDMs. We also consider a multi-scale model whose degradation matrix is a $2 \times 2$ averaging filter. In this case, unlike the diffusion process, the image is corrupted by a downsampling filter together with additive noise. Therefore, the model (termed by *RGM-SR (naive)*) is demanded to conduct upsampling and denoising at the same time. To make it more effective, we explore another forward schedule that separates the downsampling and noising process and performs them alternatively. *RGM-SR* refers to the model to which this schedule is applied. (See Appendix B.1 for details).

### 4.1 2D TOY EXAMPLE

We first employ a two-dimensional example to validate the effectiveness of prior knowledge of our MAP framework. We adopt a mixture of Gaussian with eight components (Grathwohl et al., 2018) as a target distribution. In Figure 3, we depict the benefits of our MAP approach over the MMSE approach. As illustrated in the top row, we diffuse the data distribution through four different noise levels. Each row from the second row to the bottom represents the learned distribution of our RGM, RGM without the auxiliary variable **z**, and MMSE. First, the bottom row shows the failure of MMSE, where the modes of $p_{\text{data}}$ are connected and then missed. This tendency exacerbates as the noise level increases. Since the MMSE fails to reconstruct the data distribution even with a small rise in the noise level, the MMSE does not yield a satisfactory generative model with a small number of diffusion steps.

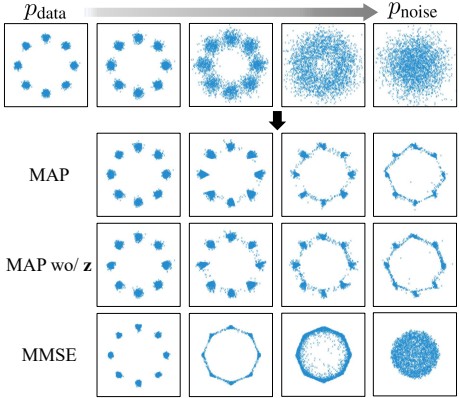

Figure 3: Comparison of recovering density by MMSE versus MAP-based objective.

Consequently, MMSE approaches, such as DDMs, require a large number of steps to stably recover the data distribution. On the other hand, by leveraging the prior term, our model generates samples

Table 1: Results on unconditional generation of CIFAR10.

| Class | Model | FID ($\downarrow$) | IS ($\uparrow$) | NFE ($\downarrow$) |
|---|---|---|---|---|
| **RGM** | RGM-D | 3.08 | 9.14 | 4 |
| | RGM-SR | 2.47 | 9.68 | 7 |
| **DDM** | DDPM (Ho et al., 2020) | 3.21 | 9.46 | 1000 |
| | NCSN (Song & Ermon, 2019) | 25.3 | 8.87 | 1000 |
| | Score SDE (VE) (Song et al., 2020b) | 2.20 | 9.89 | 2000 |
| | Score SDE (VP) (Song et al., 2020b) | 2.41 | 9.68 | 2000 |
| | Probability Flow (VP) (Song et al., 2020b) | 3.08 | 9.83 | 140 |
| | LSGM (Vahdat et al., 2021) | 2.10 | 9.87 | 147 |
| | DDIM (50 steps) (Song et al., 2020a) | 4.67 | 8.78 | 50 |
| | FastDDPM (T=50) (Kong & Ping, 2021) | 3.41 | 8.98 | 50 |
| | Recovery EBM (Gao et al., 2020) | 9.58 | 8.30 | 180 |
| | VDM (Kingma et al., 2021) | 4.00 | - | 1000 |
| | UDM (Kim et al., 2021) | 2.33 | 10.1 | 2000 |
| | Gotta Go Fast (Alexia Jolicoeur-Martineau, 2021) | 2.44 | - | 1000 |
| | Subspace Diffusion (Jing et al., 2022) | 2.17 | 9.94 | $\geq 1000$ |
| | CLD (Dockhorn et al., 2021) | 2.25 | - | 2000 |
| | DEIS (Zhang & Chen, 2022) | 3.37 | 9.74 | 15 |
| | DDGAN (Xiao et al., 2021a) | 3.75 | 9.63 | 4 |
| | StyleGAN2+ES-DDPM (Lyu et al., 2022) | 5.52 | - | 101 |
| | Progressive Distillation (Salimans & Ho, 2022) | 3.00 | - | 4 |
| **GAN** | SNGAN+DGflow (Ansari et al., 2020) | 9.62 | 9.35 | 25 |
| | AutoGAN (Gong et al., 2019) | 12.4 | 8.60 | 1 |
| | TransGAN (Jiang et al., 2021) | 9.26 | 9.02 | 1 |
| | StyleGAN2 w/o ADA (Karras et al., 2020) | 8.32 | 9.18 | 1 |
| | StyleGAN2 w/ ADA (Karras et al., 2020) | 2.92 | 9.83 | 1 |
| **VAE** | NVAE (Vahdat & Kautz, 2020) | 23.5 | 7.18 | 1 |
| | Glow (Kingma & Dhariwal, 2018) | 48.9 | 3.92 | 1 |
| | PixelCNN (Van Oord et al., 2016) | 65.9 | 4.60 | 1024 |
| | VAEBM (Xiao et al., 2020) | 12.2 | 8.43 | 16 |

from the multimodal distribution significantly better. By imposing the prior knowledge, the MAP-based objective accurately estimates the density, which allows distribution recovery with a much smaller number of forward processes than the MMSE approach. Furthermore, we can observe the effect of the auxiliary variable **z** by comparing the second and third rows in Figure 3. MAP with **z** has higher sample quality. The effect of **z** is further amplified for more complex distributions, such as image data (See Figure 5 and Table 3). This synthetic experiment validates that our MAP approach in conjunction with the random auxiliary variable **z** enables accurate and efficient generative modeling.

## 4.2 IMAGE GENERATION

We compare the performance of our method with several existing baselines. For quantitative comparison, we use Fréchet Inception Distance (FID) and Inception Score (IS) as the evaluation metrics. We also report the number of network function evaluations (NFE). For DDMs and RGMs, NFE value and real inference time are proportional. Following Song et al. (2020b); Dockhorn et al. (2021), we focus on the widely used CIFAR10 unconditional image generation benchmark and also validate the performance of RGMs on large-scale CelebA-HQ-256 images. Table 1 summarizes the quantitative evaluations on CIFAR10, and the results on CelebA-HQ is reported in Table 2. Qualitative performance is depicted in Figures 2 and 4.

**Results**   We can see that our model achieves the state-of-the-art FID score on CelebA-HQ-256 among restoraion-based models. On CIFAR10, our RGMs are comparable to the best existing DDMs and GAN models. Although the best denoising models obtain better results than ours on CIFAR10, they use a much larger number of denoising steps (e.g. ScoreSDE with VESDE requires 2000 steps). Progressive distillation achieves an FID score comparable to ours with quite reduced sampling cost, but they require considerable additional cost at training time due to their distillation process. Notably, our RGM-SR achieves FID 2.47 and IS 9.68 with only seven steps, which is state-of-the-art sampling FID performance when NFE is limited. The overall results confirm that our MAP-based estimation immediately eliminates the need for an expensive sampling scheme while still maintaining the density estimating capability of DDMs. Interestingly, RGM-SR outperforms RGM-D by a large margin even with far fewer latent variables than RGM-D. This improved performance may be attributed to the increase in NFE; however, the FID of RGM-D with $T = 8$ reported in Table 7 confirms that it is

Table 2: Results on generation of CelebA-HQ-256.

| Class | Model | FID (↓) | NFE (↓) |
|---|---|---|---|
| **RGM** | RGM-D | 7.15 | 4 |
| **DDM** | Score SDE (VP) (Song et al., 2020b) | 7.23 | 4000 |
| | Probability Flow (Song et al., 2020b) | 128.13 | 335 |
| | LSGM (Vahdat et al., 2021) | 7.22 | 23 |
| | UDM (Kim et al., 2021) | 7.16 | 2000 |
| | DDGAN (Xiao et al., 2021a) | 7.64 | 4 |
| **GAN** | PGGAN (Karras et al., 2017a) | 8.03 | 1 |
| | Adv. LAE (Pidhorskyi et al., 2020) | 19.2 | 1 |
| | VQ-GAN (Esser et al., 2021) | 10.2 | 1 |
| | DC-AE (Parmar et al., 2021) | 15.8 | 1 |
| | StyleSwin (Zhang et al., 2022) | 3.25 | 1 |
| **VAE** | NVAE (Vahdat & Kautz, 2020) | 29.7 | 1 |
| | VAEBM (Xiao et al., 2020) | 20.4 | 1 |
| | NCP-VAE (Aneja et al., 2021) | 24.8 | 1 |

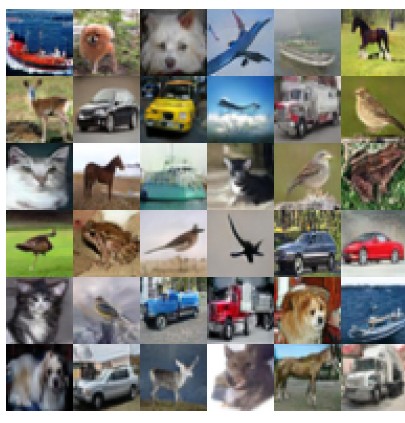

Figure 4: CIFAR10 generated samples.

not. The overall results indicate that the MAP approach of RGMs is a promising way for generating high-quality samples in limited steps. More uncurated images can be founded in Appendix C.6.

## 4.3 ABLATION STUDIES

This section is devoted to ablation analyses which show that all parts of our objective, the data fidelity term, and the prior term together coupled with the auxiliary variable, and the regularization parameter, each play an important role in our performance of density estimation.

**On the role of z** We include experimental results on LSUN, which demonstrate how the auxilliary variable z alleviates the ill-posedness of the inverse problem. By noising the upper-left image $x_0$, we obtain the forward trajectory $\{x_k\}_{k=1}^4$. The figures on the right are restored images of $x_k$ by RGM-D together with four different z. We can see that a reconstruction is almost unique when the noise level is small. But, as the noise level increases, a single $x_k$ has various reconstructions. It is evident that assigning z helps generate different denoised images from a heavily degraded $x_k$ through the guidance provided by z. However, one might think that the ill-posedness is detoured by multi-step training using multiple $\sigma_k$ rather than through z. This claim can be refuted using the result

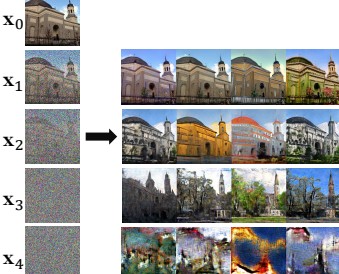

Figure 5: Study on effect of z.

of RGM-D without z reported in Table 3. We observe the significant difference in FIDs of RGM-D with and without z under the same number of denoising steps, which indicates the effectiveness of z.

**On the effect of Varying λ** We investigate the sensitivity of the regularization parameter $\lambda$ in (8). Since it controls the relative importance between the fidelity term and the prior term, $\lambda$ is a trade-off hyperparameter that determines how much regularizes the joint distribution of $p_k$ and $p_{k+1}$. In Figure 6, we present FID scores measured on CIFAR10 with the same number of degradation steps ($T = 4$) and varying $\lambda$. We can see that our models are quite robust with respect to $\lambda$. An empirically observed sweet spot of $\lambda$ is $d/10 \leq \lambda \leq d$ for the image size $d$, in which FID is no longer improved outside this threshold. For small $\lambda$, the

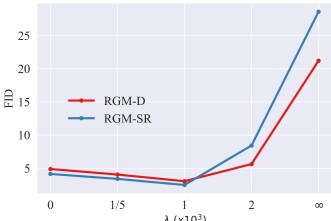

Figure 6: Study on effect of $\lambda$.

models put a lot of effort to recover the degradation, which hinders estimating data distribution. Choosing a large $\lambda$ also results in performance degeneration. There is another point that draws our attention. When $\lambda = \infty$, that is, when there is no fidelity term on the objective, FID scores completely deteriorate. In this case, the models are trained by the vanilla GAN loss. From the perspective of GAN, our MAP-based objective adds the fidelity term to the GAN loss function. We further observe that with the help of the fidelity term, our model enhances the mode-collapsing resiliency of GAN (See Figure 16). These results validate that the performance of RGMs owes to both fidelity and prior term, and the reliable regularization parameter is determined to balance these two terms.

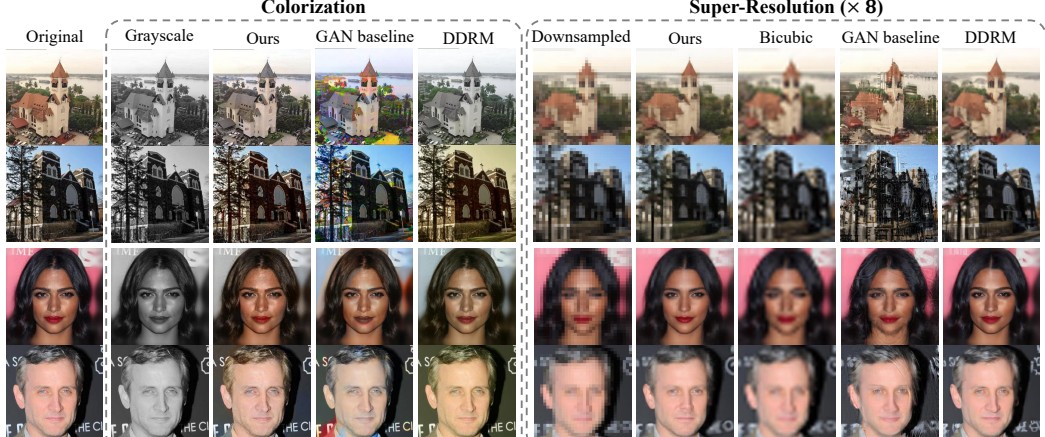

Figure 7: Colorization (left) and super-resolution (right) results on LSUN and CelebA-HQ datasets.

**On the forward process schedule** Since the forward process determines the way of connecting the data and latent distributions, it significantly affects the performance of models. The first important factor is the number of forward steps $T$, which is directly related to NFE. In Table 3, we ablate the effect of $T$. When $T = 1$, it may be difficult for the model to directly approximate the data distribution from the Gaussian noise. This is reflected in the poor FID score. We also study the forward process schedule of the SR model. We can observe that the separation of the same forward process into two steps makes the model easier to learn, and this brings the performance enhancement of RGM-SR compared to RGM-SR (naive).

Table 3: Ablation Studies.

| Model | FID ($\downarrow$) |
|---|---|
| RGM-D ($\lambda = \infty$) | 32.5 |
| RGM-D ($T = 1$) | 14.6 |
| RGM-D ($T = 4$) | 3.04 |
| RGM-D (wo / z) | 3.87 |
| RGM-SR (naive) | 3.17 |
| RGM-SR | 2.47 |

## 4.4 INVERSE PROBLEMS

While our model was originally devised to generate images, we further show the applicability of RGMs to inverse problems. Recently, a promising approach in imaging inverse problems is to leverage a learned denoiser as an alternative to the proximal operator of splitting algorithms (Romano et al., 2017; Hurault et al., 2021). Such methodology is referred to as Plug-and-Play (PnP) algorithms (Venkatakrishnan et al., 2013). In a similar spirit, we utilize the trained RGMs as a modular part of the PnP algorithms to solve various inverse problems. In this section, we testify our RGM-D for two inverse problems; super-resolution and colorization, by plugging our model into Douglas-Rachford Splitting algorithm (Lions & Mercier, 1979). Details can be found in Appendix B.3.

**Results** We compare the performance of our model with current-leading models: We compare our model with DDRM (Kawar et al., 2022), which solves inverse problems with a pre-trained DDPM by a posterior sampling scheme. As a GAN baseline, we adopt StyleSwin (Zhang et al., 2022) and reconstruct the image by optimizing over the latent vector (Pan et al., 2021). We also consider bicubic interpolation as a baseline for super-resolution. We observe that our model is capable of reconstructing faithful and realistic images, as evident in Figure 7. Compared with baselines, our model produces high-quality reconstructions across all the datasets. In particular, our model shows promising performance for colorization. These results show the applicability of RGMs to PnP prior, and this will bring a range of potential applications, including image segmentation, conditional generation, and other imaging inverse problems. Additional quantitative and qualitative results are provided in Appendix C.4.

**Comparison of RGM-D and RGM-SR** We investigate the effect of the degradation process used during training on the performance of solving inverse problems. We compare the reconstruction performance of RGM-D and RGM-SR that are trained on different degradation processes by applying both models to denoising and super-resolution (SR) tasks on CIFAR10. Quantitative results are presented in Table 4. We can see that the RGM-SR that is trained based on SR actually performs the SR task better. Also, we can observe a similar tendency for denoising. The results confirm that the degradation process used in training actually helps in solving the corresponding inverse problem.

Table 4: Quantitative comparison of RGM-D and RGM-SR on image reconstruction.

| Model | Super-Resolution | | | | Denoising | | | | | |
|---|---|---|---|---|---|---|---|---|---|---|
| | (×2) | | (×4) | | ($\sigma = 10/255$) | | ($\sigma = 20/255$) | | ($\sigma = 40/255$) | |
| | PSNR | SSIM | PSNR | SSIM | PSNR | SSIM | PSNR | SSIM | PSNR | SSIM |
| RGM-D | 26.63 | 0.88 | 20.84 | 0.58 | **30.11** | **0.93** | **26.57** | **0.86** | **24.23** | **0.80** |
| RGM-SR | **27.42** | **0.90** | **21.14** | **0.59** | 29.41 | 0.92 | 25.87 | 0.84 | 23.53 | 0.77 |

## 5 RELATED WORK

In recent years, denoising-based generative models (Ho et al., 2020; Song & Ermon, 2019; Song et al., 2020b) have emerged as a class of density estimation models, first sparked by (Sohl-Dickstein et al., 2015). They define a sampling process as the reverse of a forward diffusion process that maps data to Gaussian noise by consecutively adding a small portion of the noise to the input data. As they faithfully estimate the data distribution and generate high-fidelity samples, they have rapidly been applied to various domains such as conditional generation (Lee et al., 2022; Ho et al., 2022a), audio synthesis(Kong et al., 2021; Popov et al., 2021), medical imaging (Song et al., 2021; Chung & Ye, 2022), video generation (Ho et al., 2022b; Yang et al., 2022), and 3D point cloud generation (Lyu et al., 2021). Their major drawback is slow and expensive inference. Many studies have been dedicated to circumventing this downside by developing a fast numerical solver (Jolicoeur-Martineau et al., 2021; Zhang & Chen, 2022; Tachibana et al., 2021; Liu et al., 2022) or using an alternative noising process such as non-Markovian (Song et al., 2020a), a second-order Langevin dynamics (Dockhorn et al., 2021), and non-linear diffusion processes (De Bortoli et al., 2021; Chen et al., 2021b). Another line of work improves sampling efficiency by incorporating it into other generative models, including GAN (Xiao et al., 2021a; Lyu et al., 2022), and VAE (Vahdat & Kautz, 2020). Xiao et al. (2021a) which enjoys small sampling steps by using GAN is one of our related work. However, they did not introduce GAN from a MAP perspective and our model requires less training iteration to achieve the same performance. (See Appendix A for details.) Moreover, there have been distillation approaches (Salimans & Ho, 2022; Meng et al., 2022). On a side note, all the aforementioned models use the Gaussian noising process as the forward process.

Recently, the literature has begun to replace the additive Gaussian noising process with other transforms. Breaking away from the diffusion process, (Rissanen et al., 2022) proposed a forward blurring process inspired by heat dissipation. They suggest a new generation process, but they specialize in the proposed blurring process and cannot be incompatible with other degradation processes. Possibly the closest study to our work is Cold Diffusion (Bansal et al., 2022) which generalizes the diffusion process to arbitrary image transformations. It seems to use a general transform similar to our models, but Cold Diffusion only uses deterministic degradation processes by entirely removing additive Gaussian noise, which hinders its density estimation performance. Also, they use the MMSE objective, still requiring an array of several forward steps. We include a comparison with these related works in Appendix C.3.

## 6 CONCLUSION AND FUTURE WORK

In this study, we presented a general framework for modeling efficient generative models through the lens of IR. Compared to DDMs whose both forward and reverse processes are fixed to thousands of Gaussian steps, our approach provides more flexible models that eliminate expensive sampling and can enjoy versatile forward processes. We eliminated the usage of slow sampling by taking on MAP approach and incorporating implicit prior information through GAN. In addition,we propose a multi-scale method as an example of the usability of various forward processes. The experimental results showed that the image quality obtained was on par with the leading DDMs, and we achieved state-or-the-art performance using a limited number of forward steps. We hope that this work provides a broad view of modeling useful generative models.

Our model has two degrees of freedom: One is how to parametrize the prior knowledge, and the other is the choice of the forward process. This opens up interesting directions for future research. In addition to the GAN we used, the prior term can be constructed in different ways. It could also be interesting to explore other degradation transformations. Moreover, analyzing the effect of the random auxiliary variable would be an worthwhile direction for future research. Future work could include the comprehensive design of a convergence guaranteed PnP algorithm for application to various inverse problems. We leave these further extensions to future work.

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

## A  MORE RELATED WORKS

DDMs have been pertinent generative models by showing promising results on various generation tasks. DDMs degrade the data with a reference diffusion process and learn the data distribution by restoring it. We have arranged DDMs in the context of restoration, and DDMs can be interpreted as an MMSE estimator for a denoising task.

- Energy-based models (EBMs) are another line of generative models that learn the unnormalized data distribution by giving low energy to high-density regions in the data space. As DDMs have demonstrated that recovery of sequence of noisy data is more effective than directly approximating the data density, Gao et al. (2020) recently proposed a recovery energy-based model (REBM) by using a diffusion process. Inspired by DDMs, REBM learns a sequence of energy functions for the marginal distributions of the diffusion process. More precisely, from the noisy observation $\tilde{\mathbf{x}} = \mathbf{x} + \xi, \xi \sim \mathcal{N}\left(0, \sigma^2 I\right)$, they estimate the conditional likelihood $p_\theta\left(\mathbf{x} \mid \tilde{\mathbf{x}}\right) \propto \exp^{-\mathcal{E}_\theta(\mathbf{x}|\tilde{\mathbf{x}})}$ by learning the energy function

$$\mathcal{E}_\theta = \frac{1}{2\sigma^2}\|\mathbf{x} - \tilde{\mathbf{x}}\|^2 - f_\theta\left(\mathbf{x}\right). \tag{11}$$

They indeed learn the marginal density $f_\theta$ and inference the data through the recovery likelihood. The marginal density $f_\theta$ is adversarially trained by assigning low energy to high-probability regions in the data space and high energy values outside these regions. Since direct sampling from $p_\theta\left(\mathbf{x} \mid \tilde{\mathbf{x}}\right)$ is intractable, samples are usually drawn by leveraging Langevin dynamics (LD) (Neal, 1993), which is a conventional sampling method of EBMs. Therefore, REBM trains marginal density $f_\theta$ using a kind of adversarial loss, but REBM is actually a MAP estimator implicitly defined by the sampling dynamics. In other words, REBM learns the posterior distribution using the reference diffusion process, but it does not deviate from the traditional sampling method of EBM, still generating samples through inefficient LD. There are two difficulties of such a Markov Chain Monte Carlo (MCMC) sampling: Applying MCMC in pixel space to sample one instance from the model is impractical due to the high dimensionality and long inference time. As reported in (Xiao et al., 2021b), the estimated density of EBMs can sometimes differ significantly from the data distribution, even if the model with the short-run LD produces relevant samples. It is also known that the convergence of LD is very difficult when the energy function is complicated.

- Another related work is a denoising diffusion GAN (DDGAN) (Xiao et al., 2021a), which enjoys small sampling steps by using GAN. DDGAN focuses on improving the sampling efficiency while maintaining the sample quality and mode coverage of DDMs. The reason why DDMs adhere to the heavy sampling scheme is their common assumption that the true posterior is approximated by Gaussian distributions. This assumption holds only with small denoising steps. When the number of denoising steps is reduced, the denoising distribution is no longer a Gaussian distribution, but a non-Gaussian multi-modal, which is usually intractable. DDGAN breaks the Gaussian assumption by reducing the number of denoising steps, and then approximates the non-Gaussian multimodal posterior distribution with the help of GAN. DDGAN enhances the sampling efficiency of DDMs and also resolves the mode collapse problem of GANs by using a couple of denoising steps from the perspective of GAN literature. The architecture of DDGAN is somewhat similar to that of our RGM-D. However, there is a difference in a way of estimating MAP. DDGAN assigns all responsibility for MAP estimation to the discriminator. On the other hand, our models learn the MAP-based estimator by separating the posterior distribution into the fidelity term and the prior term. Therefore, the model is much easier to learn than DDGAN. As a consequence, RGM-D obtains substantial savings in terms of training iterations than DDGAN. Specifically, in CIFAR10 experiments, DDGAN takes 400K iterations to achieve FID of 3.75. In comparison, our RGM-D only uses 150K iterations to achieve the same performance as DDGAN, and takes 200K iterations for FID of 3.08. For the CelebA dataset, DDGAN requires 750K iterations to attain FID 7.64, while RGM-D obtains the same FID score using only 450K iterations and FID 7.15 even with much less 500K iterations.

As such, there have been various density estimation models based on denoising. Diffusion models, such as DDPM and score matching with Langevin dynamics and its variants, are MMSE-based estimators. The model of REBM itself approximates the marginal density as we do, but our model is trained with MAP-based loss, whereas REBM generates samples from the posterior distribution through the sampling method. Diffusion models and REBM train different estimators, but both

models use a Langevin sampling scheme that requires thousands of network evaluations. On the other hand, DDGAN is a model that can perform one-shot sampling with the help of GAN (away from the Langevin sampling), just like our RGMs. However, since DDGAN learns the whole posterior density through the discriminator, it is more inefficient in terms of learning than our models, which separate the fidelity and the prior term. Consequently, our RGMs achieve better performance than DDGAN with much fewer iterations. All these models are restricted to the diffusion process. Otherwise, our RGMs can enjoy flexible forward processes and are also given a degree of freedom in how to parametrize the prior term. In other words, our approach does not need to restrict to the diffusion process and unlike DDGAN, which is limited to the GAN structure, it is possible to design the prior term by leveraging different generation models. This is further discussed in Appendix C.2.

## B  IMPLEMENTATION DETAILS

### B.1  DEGRADATION SCHEDULE

Let $\mathbf{A}_k$ and $\mathbf{\Sigma}_k$ be a degradation matrix and a noise variance on the $k$-th degradation step, respectively. Then, given a data $\mathbf{x}$ sampled from the real data distribution $p_{\text{data}}$, a degraded data $\mathbf{y}_k$ on the $k$-th forward step is sampled from

$$p\left(\mathbf{y}_k \mid \mathbf{x}\right) = \mathcal{N}\left(\mathbf{y}_k; \mathbf{A}_k \mathbf{x}, \mathbf{\Sigma}_k\right).$$

We denote the marginal distribution at the $T$-th degradation step as $p_T$. Because our primary goal is to bridge $p_{\text{data}}$ to an easy to sample distribution $p_T$, (especially to a zero mean Gaussian distribution), we gradually decrease the norm of $\mathbf{A}_k$ to zero as $k$ increases. In Section 4, we introduced two families of models based on the degradation schedule $\{(\mathbf{A}_k, \mathbf{\Sigma}_k)\}_{k=1}^{T}$ with the corner cases: RGM-D for $\mathbf{A}_k = \mathbf{I}$ and RGM-SR for $\mathbf{A}_k = \mathbf{P}_k$ a $2 \times 2$ averaging filter. Roughly speaking, we consider three models based on different forward processes designed as follows:

- RGM-D: noise $\to$ noise $\to$ noise $\to$ noise $\to \cdots$,
- RGM-SR (naive): downsample + noise $\to$ downsample + noise $\to \cdots$,
- RGM-SR: noise $\to$ downsample $\to$ noise $\to$ downsample $\to \cdots$.

With the following notations

$$\beta_k = \frac{1}{4}\left(\beta_{\max} - \beta_{\min}\right)\left(\frac{k}{T}\right)^2 + \frac{1}{2}\beta_{\min}\frac{k}{T}, \tag{12}$$

$$\tilde{\beta}_k = \frac{1}{4}\left(\beta_{\max} - \beta_{\min}\right)\left(\frac{k}{T}\right)^4 + \frac{1}{2}\beta_{\min}\left(\frac{k}{T}\right)^2, \tag{13}$$

where $\beta_{\max} = 20$ and $\beta_{\min} = 0.1$. Table 5 details the explicit form of the forward processes used for each models.

Table 5: The choice of schedule $\mathbf{A}_k$ and $\mathbf{\Sigma}_k$ and the corresponding latent distribution $p_T$ for RGM-D, RGM-SR (naive), and RGM-SR. $\mathbf{P}_k$ is a projection matrix that downscale the images by block averaging in a factor of $2^k$. For RGM-SR, we set $T$ in (12) be a half of the total steps added by one.

|  | RGM-D | RGM-SR (naive) | RGM-SR |
|---|---|---|---|
| $\mathbf{A}_k$ | $e^{-\beta_k}\mathbf{I}$ | $e^{-\tilde{\beta}_k}\mathbf{P}_k$ | $e^{-\beta_{\lceil k/2 \rceil}}\mathbf{P}_{\lfloor k/2 \rfloor}$ |
| $\mathbf{\Sigma}_k$ | $\left(1 - e^{-2\beta_k}\right)^2 \mathbf{I}$ | $\left(1 - e^{-2\tilde{\beta}_k}\right)^2 \mathbf{P}_k^\top \mathbf{P}_k$ | $\left(2^{\lceil k/2 \rceil}\left(1 - e^{-2\beta_{\lceil k/2 \rceil}}\right)\right)^2 \mathbf{P}_{\lfloor k/2 \rfloor}^\top \mathbf{P}_{\lfloor k/2 \rfloor}$ |
| $p_T$ | $\mathcal{N}\left(\mathbf{0}, \mathbf{I}\right)$ | $\mathcal{N}\left(\mathbf{0}, \frac{1}{64}\mathbf{I}\right)$ | $\mathcal{N}\left(\mathbf{0}, 4\mathbf{I}\right)$ |

The noise schedule of RGM-D follows the Variance Preserving SDE provided by Song et al. (2020b), and others are implemented with a slight modification of them.

When we use the degradation matrix $\mathbf{A}_k$ as the averaging filter, the corresponding forward process downsamples the image while adding Gaussian noise. RGM according to this forward process, referred to as RGM-SR (naive), is demanded to super-resolve the degraded data while simultaneously

denoising it. It is considerably more difficult than the denoising task when the noise level is the same. To address this difficulty, we consider a newly scheduled degradation scheme that decomposes the forward process into downsampling and noising operations. We name the RGM designed in conjunction with this forward schedule as RGM-SR. As provided in Table 5, when the step $k$ is odd, the difference from the $(k + 1)$-th step is only the projection matrix. Namley, only downsample is performed when sampling the $(k + 1)$-th degraded data from the k-th degraded observation. Conversely, when $k$ is an even number, the forward process produces the $(k + 1)$-th degraded image by adding the Gaussian noise. In summary, RGM-SR focuses on denoising the data in odd steps and super-resolving the data in even steps. Provably due to the difficulty of performing super-resolution and denoising simultaneouly, RGM-SR (naive) has the worst performance. Whereas RGM-SR, which uses the decomposed forward process, outperforms both RGM-D and RGM-SR by a large margin as reported in Section 4.2.

## B.2 TRAINING RGMs

In this section, we unambiguously elucidate how we train our RGMs. In Algorithm 1 and 2, we summarize the two training procedures that are suited to different situations. Moreover, the generation process is provided in Algorithm 3.

**Training**  As proposed in Section 3.2, RGMs learn the data distribution $p_{\text{data}}$ through the process of degrading the image through a forward process and then restoring it using the MAP-based objective (8). However, since it is too difficult to restore the image directly from the Gaussian distribution in one shot, we use a handful of forward steps and train RGMs with the MAP estimation that recovers the distribution between each step. (We also include an ablation study on this in Appendix C.1) In other words, at each step $k$, we first sample a degraded image $\mathbf{y}_k$ of a given image $\mathbf{x} \sim p_{\text{data}}$. The generator $G_\theta$ generates the restored image $\hat{\mathbf{x}}$, and then, we degrade it by the posterior distribution $\hat{\mathbf{y}}_{k-1} \sim p\left(\hat{\mathbf{y}}_{k-1} \mid \mathbf{y}_k, \hat{\mathbf{x}}\right)$. We train our MAP-based loss function so that $\hat{\mathbf{y}}_{k-1}$ becomes a restoration of $\mathbf{y}_k$. The discriminator loss is also imposed on the $(k - 1)$-th step. Through the overall process, we ultimately learn the model that restores the distribution of the previous $(k - 1)$-th step at each $k$-th step. The training procedure is articulated in Algorithm 1.

---

**Algorithm 1** Training of RGMs with Posterior sampling

**Input:** Dataset $\mathcal{D}$, degradation schedule $\{(\mathbf{A}_k, \boldsymbol{\Sigma}_k)\}_{k=0}^T$ with $(\mathbf{A}_0, \boldsymbol{\Sigma}_0) = (\mathbf{I}, \mathbf{0})$, posterior distribu-
   tion $p_{k|k-1}\left(\mathbf{y}_{k-1}, \mathbf{y}_k\right) = \mathcal{N}\left(\tilde{\mathbf{A}}_k \mathbf{y}_k, \tilde{\boldsymbol{\Sigma}}_k\right)$, generator $G_\theta$, discriminator $D_\phi$, and regularization
   parameter $\lambda \geq 0$.
1: **for** $i = 0, 1, 2, \ldots$ **do**
2:     Sample data $\mathbf{x} \in \mathcal{D}$.
3:     Sample $k \sim \text{Uniform}(\{1, 2, \ldots, T\})$.
4:     Sample $\mathbf{z} \sim \mathcal{N}(0, \mathbf{I})$.
5:     Sample degraded data $\mathbf{y}_k \sim \mathcal{N}\left(\mathbf{A}_k \mathbf{x}, \boldsymbol{\Sigma}_k\right)$ and $\mathbf{y}_{k-1} \sim \mathcal{N}\left(\mathbf{A}_{k-1} \mathbf{x}, \boldsymbol{\Sigma}_{k-1}\right)$.
6:     Generate an image $\hat{\mathbf{x}} = G_\theta(\mathbf{y}_k, k, \mathbf{z})$.
7:     Degrade data by posterior sampling $\hat{\mathbf{y}}_{k-1} \sim p\left(\hat{\mathbf{y}}_{k-1} \mid \mathbf{y}_k, \hat{\mathbf{x}}\right)$.
8:     Update $\phi$ by the following loss:

$$\log\left(1 - D_\phi\left(\hat{\mathbf{y}}_{k-1}, k - 1\right)\right) + \log D_\phi\left(\mathbf{y}_{k-1}, k - 1\right).$$

9:     Update $\theta$ by the following loss:

$$\log\left(1 - D_\phi\left(\hat{\mathbf{y}}_{k-1}, k - 1\right)\right) - \log D_\phi\left(\hat{\mathbf{y}}_{k-1}, k - 1\right) + \frac{1}{2\lambda}\left\|\left(\tilde{\boldsymbol{\Sigma}}_k^\dagger\right)^{\frac{1}{2}}\left(\tilde{\mathbf{A}}_k \hat{\mathbf{y}}_{k-1} - \mathbf{y}_k\right)\right\|_2^2.$$

10: **end for**

---

However, we can exactly formulate the posterior distribution only when the forward process satisfies certain conditions. For all $k = 1, \cdots, T$, if there exists $\left(\tilde{\mathbf{A}}_k, \tilde{\boldsymbol{\Sigma}}_k\right)$ satisfying

$$\mathbf{A}_k = \tilde{\mathbf{A}}_k \mathbf{A}_{k-1}, \ \tilde{\boldsymbol{\Sigma}}_k := \boldsymbol{\Sigma}_k - \tilde{\mathbf{A}}_k \boldsymbol{\Sigma}_k \tilde{\mathbf{A}}_k^\top \succ 0, \tag{14}$$

we can explicitly construct a conditional distribution $p_{k|k-1}(\mathbf{y}_k|\mathbf{y}_{k-1}) = \mathcal{N}(\tilde{\mathbf{A}}_k\mathbf{y}_{k-1}, \tilde{\mathbf{\Sigma}}_k)$ and a posterior distribution (Ho et al., 2020; Kingma et al., 2021; Xiao et al., 2021a). For example, the forward process of RGM-D falls under this condition (14), but that of RGM-SR does not. Therefore, the Algorithm 1 does not fit with RGM-SR. To unravel such a problem, we propose a prevalent algorithm that is applicable to forward processes that are in discord with the condition (14). See Algorithm 2. The only difference from the Algorithm 1 is the replacement of the posterior sampling by the prior sampling in Line 5 and the data fidelity term in Line 9. When the posterior distribution is unavailable, we corrupt the image $\hat{\mathbf{x}}$ restored by the generator $G_\theta$ to the $(k-1)$-th degraded distribution using the $k$-th forward process rather than posterior sampling. Moreover, since the conditional distribution between $k$ and $(k-1)$ steps is unknown, we adopt the fidelity term of the image $\hat{\mathbf{x}}$ reconstructed by the generator. This algorithm is universally applicable to general forward processes. One notable fact is that RGM-D, whose posterior distribution is tractable, learns the data distribution better when using this algorithm than Algorithm 1. This is discussed in detail in Appendix C.1.

---

**Algorithm 2** Relaxed training algorithm of RGMs

---

**Input:** Dataset $\mathcal{D}$, degradation schedule $\{(\mathbf{A}_k, \mathbf{\Sigma}_k)\}_{k=0}^T$ with $(\mathbf{A}_0, \mathbf{\Sigma}_0) = (\mathbf{I}, \mathbf{0})$, discriminator $D_\phi$, generator $G_\theta$, and regularization parameter $\lambda \geq 0$.

1: **for** $i = 0, 1, 2, \ldots$ **do**
2:     Sample $\mathbf{x} \in \mathcal{D}$.
3:     Sample $k \sim \text{Uniform}(\{1, 2, \ldots, T\})$.
4:     Sample $\mathbf{z} \sim \mathcal{N}(0, \mathbf{I})$.
5:     Sample degraded data $\mathbf{y}_k \sim \mathcal{N}(\mathbf{A}_k\mathbf{x}, \mathbf{\Sigma}_k)$ and $\mathbf{y}_{k-1} \sim \mathcal{N}(\mathbf{A}_{k-1}\mathbf{x}, \mathbf{\Sigma}_{k-1})$.
6:     Generate an image $\hat{\mathbf{x}} = G_\theta(\mathbf{y}_k, k, \mathbf{z})$.
7:     Degrade $\hat{\mathbf{x}}$ by $\hat{\mathbf{y}}_{k-1} \sim \mathcal{N}(\mathbf{A}_{k-1}\hat{\mathbf{x}}, \mathbf{\Sigma}_{k-1})$.
8:     Update $\phi$ by the following loss:

$$\log(1 - D_\phi(\hat{\mathbf{y}}_{k-1}, k-1)) + \log D_\phi(\mathbf{y}_{k-1}, k-1).$$

9:     Update $\theta$ by the following loss:

$$\log(1 - D_\phi(\hat{\mathbf{y}}_{k-1}, k-1)) - \log D_\phi(\hat{\mathbf{y}}_{k-1}, k-1) + \frac{1}{2\lambda}\left\|\left(\mathbf{\Sigma}_k^\dagger\right)^{\frac{1}{2}}(A_k\hat{\mathbf{x}} - \mathbf{y}_k)\right\|_2^2.$$

10: **end for**

---

**Sampling** The sampling algorithm is summarized in Algorithm 3. Starting from a latent variable $\mathbf{y}_T \sim p_T$, the trained $G_\theta$ generates the restored image $\tilde{\mathbf{x}} = G_\theta(\mathbf{y}_{k+1}, k, \mathbf{z})$ with a randomly selected auxiliary variable $\mathbf{z}$ from the $(k+1)$-the degraded image $\mathbf{y}_{k+1}$, and then corrupt it by passing the $k$-th forward process. Continue this procedure until $k = 0$. When we train our model with Algorithm 1, the line 5 should be replaced by the posterior sampling.

---

**Algorithm 3** Sampling Procedure of RGMs

---

**Input:** Trained generator $G_\theta$ and degradation schedule $\{\mathbf{A}_k, \mathbf{\Sigma}_k\}_{k=1}^T$.

1: Sample initial state $\mathbf{y}_T \sim \mathcal{N}(0, \mathbf{\Sigma}_T)$.
2: **for** $k = T - 1, T - 2, \ldots, 0$ **do**
3:     Sample $\mathbf{z} \sim \mathcal{N}(\mathbf{0}, \mathbf{I})$.
4:     Restore image $\hat{\mathbf{x}}_k$ by $\hat{\mathbf{x}}_k = G_\theta(\mathbf{y}_{k+1}, k+1, \mathbf{z})$.
5:     Sample $\mathbf{y}_k \sim \mathcal{N}(\mathbf{A}_k\hat{\mathbf{x}}_k, \mathbf{\Sigma}_k)$.
6: **end for**
7: **return** $\hat{\mathbf{x}}_0$

---

**Hyperparameters** To optimize our RGMs, we are mostly following the previous literature (Xiao et al., 2021a), including network architectures, $R_1$ regularization, and optimizer settings. Note that our code is largely built on top of DDGAN [1] (MIT License). We vary the discriminator by simply changing input channels into three. Moreover, we use a learning rate of $2 \times 10^{-4}$ for generator update in all experiments and a learning rate of $10^{-4}$ for discriminator update. We use $\lambda^{-1} = 10^{-3}$ for image size of 32, and $\lambda^{-1} = 5 \times 10^{-5}$ for image size of 256. The models are trained with

---

[1]`https://github.com/NVlabs/denoising-diffusion-gan`

Adam (Kingma & Ba, 2014) in all experiments. In CIFAR10 experiments, we train RGM-D and RGM-SR (naive) for 200K iterations and RGM-SR for 230K iterations. For CelebA-HQ, we train RGM-D for 500K iterations, and use 300K iterations in LSUN experiments. In the implementation of the two-dimensional Gaussian Mixture, we use 3-layered MLP of 32 hidden dimension for both generator and discriminator with Tanh activation. We concatenated all the inputs and passed through the network. They are trained for 100K iterations with a learning rate of $10^{-4}$, batch size of 1000.

**Other details** We train our models on CIFAR-10 using 4 V100 GPUs. The training takes approximately 40 hours on CIFAR-10. Moreover, the sampling 100 samples takes approximately 0.25 seconds for RGM-D on single V100 GPUs. For evaluation on CIFAR10, we use 50K generated samples to measure IS and FID. For CelebA-HQ-256, we use 30K samples to compute FID.

### B.3 SOLVING INVERSE PROBLEMS

Modern image processing algorithms reconstruct the groundtruth image by solving the following minimization problem:

$$\underset{\mathbf{x}}{\text{minimize}} \; f_{\mathbf{y}}(\mathbf{x}) + \lambda g(\mathbf{x}),$$

where $f$ measures the fidelity to a corrupted observation $\mathbf{y}$, and $g$ constrains the solution space by measuring the complexity or noisiness of the image. Many imaging inverse problems, such as colorization, super-resolution (SR), and deblurring, fall under this form. Since the above optimization problem does not have a closed-form solution in general, first-order proximal splitting algorithms, including half-quadratic splitting (HQS) (Geman & Yang, 1995), alternating direction method of multipliers (ADMM) (Boyd et al., 2011), solve the problem by operating individually on $f$ and $g$ via the proximal operator (Parikh et al., 2014). With the aid of the emergence of deep learning, Plug-and-Play (PnP) algorithms (Venkatakrishnan et al., 2013) have recently begun to connect proximal splitting algorithms and deep neural networks by replacing the proximity operator of the regularization term $g$ with a generic denoiser (Romano et al., 2017; Reehorst & Schniter, 2018).

Similarly, our trained RGMs can be used as PnP priors. In Section 4.4 we solved two inverse problems, colorization and super-resolution, by plugging the trained RGMs into Douglas-Rachford Splitting (DRS) algorithm (Lions & Mercier, 1979), following (Hurault et al., 2022). This is summarized in Algorithm 4. Starting from the degraded observation $\mathbf{y}$, the DRS algorithm updates the solution by alternatively utilizing proximal operations for both $f$ and $g$. By iteratively updating the solution, the solution lies far outside the distribution on which our denoiser $G_\theta$ trained. For this out-distribution data, $G_\theta$ cannot recover the original image distribution, which in turn prevents the DRS algorithm from convergence. To remedy this problem, the input of $G_\theta$ should always be within the trained distribution. Therefore, we push the updated solution into the learned distribution through the forward process. Note that the proximal operation is calculated by utilizing efficient singular value decomposition proposed in Kawar et al. (2022).

---

**Algorithm 4** Solving Inverse Problems by RGM

---

**Input:** A degraded observation $\mathbf{y}$, fidelity loss function $f_{\mathbf{y}}$, repeat number $M$, update rate $\alpha \in (0, 1]$, regularization parameter $\lambda \geq 0$, trained generator $G_\theta$, and degradation schedule $\{\mathbf{A}_k, \mathbf{\Sigma}_k\}_{i=1}^{T}$.

1:  $\mathbf{x}_K = \mathbf{y}$
2:  **for** $0, 1, \ldots, M$ **do**
3:      **for** $i = K, K-1, \ldots, 1$ **do**
4:          Sample $\hat{\mathbf{y}} \sim \mathcal{N}(\mathbf{A}\mathbf{x}_i, \mathbf{\Sigma}_i)$ and $\mathbf{z} \sim \mathcal{N}(0, \mathbf{I})$.
5:          $\hat{\mathbf{x}} \leftarrow G_\theta(\hat{\mathbf{y}}, i-1, \mathbf{z})$.
6:          $\hat{\mathbf{x}} \leftarrow (1-\alpha)\mathbf{x}_i + \alpha\hat{\mathbf{x}}$.
7:          $\Delta\mathbf{x} \leftarrow \text{prox}_{\lambda f_{\mathbf{y}}}(2\hat{\mathbf{x}} - \mathbf{x}_i) - \hat{\mathbf{x}}$.
8:          $\mathbf{x}_{i-1} \leftarrow \mathbf{x}_i + \Delta\mathbf{x}$.
9:      **end for**
10: **end for**
11: **return** $\mathbf{x}_0$

---

**Settings & Hyperparameters** In SR experiments, we downscale images by using a block averaging filter by $r$ in each axis. The filter is applied for stride of $r$. We experiment on $r = 4$ and $r = 8$ for

LSUN and CelebA-HQ datasets. In CIFAR10 experiment, we use $r = 2$ and $r = 4$. In colorization experiments, we simply degrade color images to gray by averaging images along channels of each pixel. All tasks are evaluated on hundred samples which are sampled from evaluation dataset.

Table 6 reports the exact set of hyperparameters that we used in our experiments. We set $K = 2$ for colorization and $K = 1$ for denoising and SR tasks.

On CIFAR10 experiments, to fairly compare RGM-D and vanilla version of RGM-SR, we train both models with same degradation steps of three ($T = 3$). For RGM-D, we used $\mathbf{A}_k = e^{-\tilde{\beta}_k} \mathbf{I}$ and $\boldsymbol{\Sigma}_k = \left(1 - e^{-2\tilde{\beta}_k}\right)^2 \mathbf{I}$. For RGM-SR, we used $\mathbf{A}_k = e^{-\tilde{\beta}_k} \mathbf{P}_k$ and $\boldsymbol{\Sigma}_k = \left(1 - e^{-2\tilde{\beta}_k}\right)^2 \mathbf{P}_k^\top \mathbf{P}_k$.

Table 6: Hyperparameters used for solving inverse problems.

| | CIFAR10 | | | | | LSUN/CelebA-HQ | | |
| | SR2 | SR4 | $\sigma = 10/255$ | $\sigma = 20/255$ | $\sigma = 40/255$ | SR4 | SR8 | Color |
|---|---|---|---|---|---|---|---|---|
| M | 5 | 10 | 10 | 20 | 10 | 40 | 40 | 20 |
| $\lambda$ | 0.2 | 0.1 | 0.01 | 5 | 5 | 10 | 10 | 5 |
| $\alpha$ | 0.2 | 0.2 | 0.2 | 0.1 | 0.1 | 0.05 | 0.05 | 0.5 |

**Baselines**  We employed two main comparison models, namely DDRM (Kawar et al., 2022) and GAN baseline, which is close to our work. Similar to our method, both comparison models assume that a degradation matrix is given and they iteratively update degraded images by using their knowledge obtained from pretrained network and degradation matrix. Moreover, our model and these comparisons does not require heavy additional training. The implementation DDRM follows its original implementation. The implementation of GAN baseline mainly follows the implementation of DGP (Pan et al., 2021), however, instead of using BigGAN (Brock et al., 2019), we replaced it with a pretrained model of StyleSwin (Zhang et al., 2022), which is one of the state-of-the-art. For discriminator loss of DGP, we used last feature vector of StyleSwin discriminator. We additionally adjusted the weights of the losses. For experiments in SR, we use MSE loss weight of 1.0 and discriminator loss weight of 1.0. For colorization, we use MSE loss weight of 1.0 and discriminator loss of 1.0 for previous 400 iterations and 0.1 after that. Other hyperparameters of GAN baseline implementation follows Pan et al. (2021). We also compare our model with SDEdit Meng et al. (2021), a stroke-based diffusion model. In the implementation of SDEdit, we use total denoising steps of 200 with the number of repeats of three.

# C  ADDITIONAL RESULTS

## C.1  ADDITIONAL ABLATION STUDIES

In this section, we include additional ablation studies on our training procedure and the forward process schedule. All experiments are conducted on the CIFAR10 dataset and focused on RGM-D.

**Directly restoring the data distribution**  Given a $k$-th degraded image $\mathbf{y}_k$, the generator is trained to restore the original image in one shot. Therefore, we can train RGMs to directly restore the real image distribution from each degraded step $k$. RGM-D trained in this say is denoted by *Directly matching data* in the Table 7. This model was trained in the same forward process as RGM-D ($T = 4$). The FID score shows that the model has difficulties in learning the data distribution, falling short of FID score by 21.2. It seems that it is still difficult to directly restore the image of the real data distribution from a severely degraded image $\mathbf{y}_k$ ($k \approx T$) even with the help of auxiliary variable $\mathbf{z}$.

Table 7: Additional ablation studies on CIFAR10 experiments.

| Model | FID ($\downarrow$) |
|---|---|
| Directly matching data | 21.2 |
| RGM-D w/ posterior | 3.52 |
| RGM-D ($T = 8$) | 6.50 |
| RGM-D ($T = 4$) | 3.04 |

**Training with posterior sampling**  As introduced in B.2, there are two ways to push the image, reconstructed by the generator, to the $(k-1)$-th degraded distribution; prior sampling and posterior sampling. The posterior sampling is theoretically well-grounded since it minimizes the statistical MAP loss of the posterior distribution. However, to obtain an explicit form of posterior sampling, the forward process should be constrained to satisfy the conditions (14). Since the noising forward

process of RGM-D satisfies these conditions, we trained RGM-D with both posterior sampling (Algorithm 1) and prior sampling (Algorithm 2) under the same setup. In Table 7, *RGM-D (T = 4)* and *RGM-D w/ posterior* refer the model trained with prior and posterior sampling, respectively. As shown in Table 7, both models achieve similar results in terms of FID score, where RGM-D with prior sampling slightly precedes posterior sampling. This verifies that two training objectives of Algorithm 1 and 2 are somewhat consistent. Because the performance is a bit better, we adopt the prior sampling in all our experimental studies.

**Effect of the number of forward steps**    The number of forward steps is one of the important factors affecting the performance of the model. We investigated this in Section 4.3 by comparing a four-step model RGM-D ($T = 4$) with the RGM-D ($T = 1$), where we use only one degradation step. As reported in Table 3, RGM-D ($T = 1$) struggles to learn the data distribution because it needs to recover the real data distribution directly from Gaussian noise with one chance. On the other hand, RGM-D ($T = 4$) estimates the data density well. Besides, what happens when we use more steps? Since our RGMs learn the data distribution in a way that restores the distribution of the previous degradation step ($k - 1$) distribution from the $k$-th degraded distribution, one may expect that the models will be easier to estimate the density as the distribution between the two steps is closer by dividing the forward process with more steps. However, the opposite results are presented in the Table 7. The results show that RGM-D ($T = 8$) attains a higher FID score. In other words, dividing the forward process into smaller pieces does not enhance the model performance. In addition, this phenomenon is also observed for Directly matching data. RGM-D ($T = 1$) can actually be regarded as a Directly matching data ($T = 1$), whereas the Directly matching data presented in the table uses $T = 4$. Comparing these two, we can observe that that the model using fewer degradation steps performs better. Xiao et al. (2021a) reported a similar tendency. Choosing appropriate $T$ is crucial for algorithmic performance, but not straightforward how many steps are optimal.

**Reducing mode collapse using data fidelity**    Lastly, we examine the influence of the data fidelity term in our MAP-based estimation. To quantify the contribution of the fidelity term, we trained RGM-D by the loss function without the data fidelity loss (termed by RGM-D ($\lambda = \infty$)) in Section 4.3, and we reached an FID score of 32.5 (See Table 3). This result clearly motivates our objective. Moreover, we observe the mode collapse for RGM-D ($\lambda = \infty$), which is the one of common failure modes of GAN. As evidence, generated samples are presented in Figure 16. Comparing samples generated by our RGM-D (see Figure 2) to Figure 16, it is clear that images generated by RGM-D have higher diversity and better quality. The results verify that it is beneficial to train our RGMs together with the data fidelity term.

## C.2    EXPERIMENTS ON THE FLEXIBILITY OF THE PRIOR TERM

Without being tied by the GAN structure presented in Section 3.2, our RGMs have the freedom to parametrize the prior term $g$ of regularizer (8) in any other way. To demonstrate that RGM framework universally works for variously parametrized prior terms, we design the prior term in two additional ways: *maximum mean discrepancy (MMD)* (Dziugaite et al., 2015) and *distributed sliced Wasserstein distance (DSWD)* (Nguyen et al., 2020):

- Different to GAN, the MMD-based generative model replaces the discriminator in GAN with a two-sample test based on kernel maximum mean discrepancy (MMD) (Li et al., 2017). For given two sets of data $X = \{x_1, x_2 \dots, x_M\}$ and $Y = \{y_1, y_2 \dots, y_M\}$, the MMD prior $g(X, Y)$, which estimates the MMD distance, is defined as follows;

$$g\left(X, Y\right) = \frac{1}{\binom{M}{2}} \left[ \sum_{i \neq j} k\left(x_i, x_j\right) - 2 \sum_{i \neq j} k\left(x_i, y_j\right) + \sum_{i \neq j} k\left(y_i, y_j\right) \right], \qquad (15)$$

where $k$ is a positive definite kernel. Following the prior works (Dziugaite et al., 2015; Li et al., 2015; 2017), we use a mixture of RBF kernels $k(x, x') = \sum_{i=1}^{n} k_{\sigma_i}(x, x')$ where $k_\sigma$ is a Gaussian kernel with bandwidth parameter of $\sigma$.

- To measure the distance of two datasets $X = \{x_1, x_2 \dots, x_M\}$ and $Y = \{y_1, y_2 \dots, y_M\}$, sliced Wasserstein-based framework (SW) projects the data into a one dimensional vector then explicitly calculates the Wasserstein distance on the projected space. In such an explicit calculation, SW

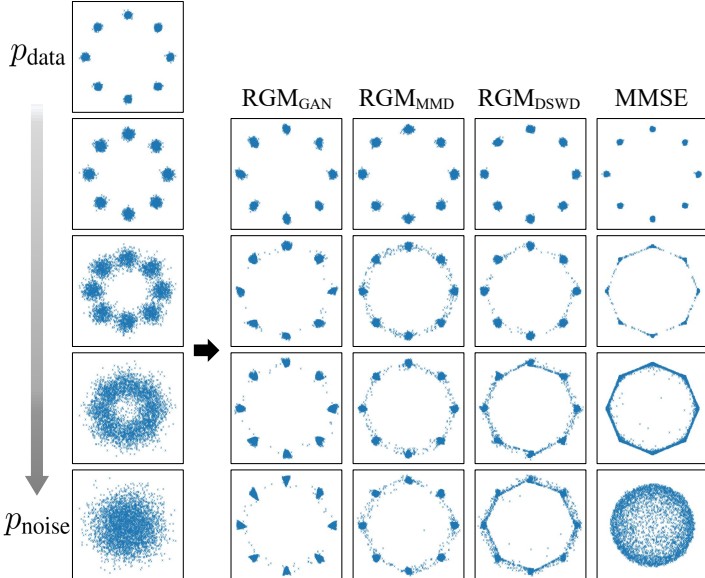

Figure 8: Efficiency of MAP-based approach compared to MMSE. The prior term of our objective is variously parametrized with GAN, MMD, and DSWD. All three MAP-based approaches are much more efficient than MMSE approach, which shows the flexibility of how to parameterize the prior term of the RGMs.

can be freed from unstable adversarial framework. Recently, Nguyen et al. (2020) has proposed a novel and efficient method to obtain the useful projection samples, hence, we followed the implementation of this prior work in our experiments.

The results on the 2D syntehtic example discussed in Section 4.1 are depicted in Figure 8. The results validate that RGMs parametrized in three different ways show consistent performance, where they are all more efficient than the MMSE estimator. In particular, MMD measures the distance between two distributions based on a kernel. Because the kernel is not trained, the prior term in the objective of RGM is fixed rather than learned like GAN. Also, despite this simple structure, Figure 3 confirms that our RGM with MMD is more efficient than MMSE.

Furthermore, we also carry out the experiment of RGM-D with the DSWD prior, termed RGM-D-DSWD, on CIFAR10. RGM-D-DSWD achieves an FID score of **3.14** retaining comparable performance with RGM-D with GAN prior. The overall results verify that our MAP approach works universally well even when the prior term is parametrized in a way other than the GAN structure.

**Implementation details** For both 2D toy and CIFAR10 experiments, we use the same architecture and hyperparameters with RGM-D unless stated. We use the number of iterations of 150K, the number of projections of 1000, 10 DSW iterations and $\lambda_C = 10$. For RGM-D-DSWD implementation on CIFAR10, we use the output of fifth convolutional layer of the discriminator as a feature vector. For the DSWD experiment on the 2D data, we use the number of iterations of 100K, the number of projections of 10, 10 DSW iterations and $\lambda_C = 10$. We refer to (Nguyen et al., 2020) for the precise definition of hyperparameters. For the MMD experiment, we applied kernel bandwidths of 0.1, 0.5, 1, 2, and 10.

### C.3 COMPARISON WITH EXISTING MODELS USING VARIOUS DESTRUCTION

Recently, several works introduce various degradation processes as an alternative to the diffusion process. Rissanen et al. (2022) proposed an inverse heat dissipation model (IHDM) with a forward blurring process inspired by heat equation. Afterwards, Hoogeboom & Salimans (2022) established a theoretical bridge between diffusion models and IHDM using Fourier transform. Based on this insight, they built a blurring diffusion model. Daras et al. (2022) proposed a general framework for learning the score function for any linear corruption process. Moreover, Cold Diffusion (Bansal et al., 2022) proposed a new family of models using deterministic degradation processes. Similarly, the proposed RGMs can leverage general linear degradation processes. Therefore, we compare the

performance of RGMs with the aforementioned related works in Table 8. In comparison with our model itself, the change in the forward process brings FID improvement. But compared to other models, we can observe how efficiently our MAP-based approach produces high-quality images.

Table 8: Comparison with restoration-based models with various forward processes. Sample quality on CIFAR10 is measured by FID score.

| Model | FID ($\downarrow$) | NFE |
|---|---|---|
| Cold Diffusion (SR) (Bansal et al., 2022) | 152.76 | 3 |
| Cold Diffusion (Blur) (Bansal et al., 2022) | 80.08 | 50 |
| IHDM (Rissanen et al., 2022) | 18.96 | 200 |
| Soft Diffusion (Daras et al., 2022) | 3.86 | $\leq 100$ |
| Soft Diffusion (Blur) (Daras et al., 2022) | 4.64 | $\leq 100$ |
| Blurring Diffusion (Hoogeboom & Salimans, 2022) | 3.17 | 1000 |
| RGM-D | 3.04 | 4 |
| RGM-SR | 2.47 | 7 |

## C.4 ADDITIONAL RESULTS ON INVERSE PROBLEMS

To quantify the performance of our RGM, we report signal-to-noise ratio (PSNR), which measures faithfulness to the ground-truth image. Also, as a perceptual metric, we include structural similarity index measure (SSIM) (Wang et al., 2004) that quantifies image. Table 9 summarizes the PSNR and SSIM performances of colorization and super-resolution (SR) on CelebA-HQ and LSUN datasets. Since the primary goal of SDEdit is to generate a realistic and faithful image in the absence of paired data, we did not make a quantitative comparison with SDEdit. But we include qualitative comparisons.

**Colorization** The goal of image colorization is to restore a gray-scale image to a colorful image with RGB channels. We present more colorization results on CelebA-HQ and LSUN church in Figure 10 and 11, respectively. Results reported in Table 9 show that our RGM achieves comparable and sometimes even better performance than baselines. From the qualitative results, we can observe that our RGM is able to reconstruct more faithful and realistic images than other models.

**Super-resolution** Super-resolution aims at recovering high-resolution image the corresponding to a given low-resolution image. We consider downsampled images with two scale scale factors 4 and 8. We also compare SR results with bicubic interpolation. Figure 12 and 13 present the qualitative comparisons. Compared against bicubic upsampling, bicubic attains higher PSNR and SSIM values. However, we can observe from Figure 12 and 13 that bicubic interpolation results in blurry images and RGM super-resolves more plausible images. Also, visual differences between RGM and DDRM are qualitatively not large.

Table 9: Colorization and super-resolution results of different methods.

| Model | Colorization | | | | Super-Resolution | | | | | | | |
|---|---|---|---|---|---|---|---|---|---|---|---|---|
| | LSUN | | CelebA-HQ | | LSUN | | | | CelebA-HQ | | | |
| | | | | | ($\times 4$) | | ($\times 8$) | | ($\times 4$) | | ($\times 8$) | |
| | PSNR | SSIM | PSNR | SSIM | PSNR | SSIM | PSNR | SSIM | PSNR | SSIM | PSNR | SSIM |
| RGM | 23.78 | 0.93 | 25.57 | 0.93 | 22.74 | 0.65 | 19.96 | 0.48 | 28.51 | 0.81 | 24.86 | 0.70 |
| DDRM | 23.68 | 0.94 | 23.94 | 0.93 | 23.22 | 0.67 | 20.61 | 0.51 | 29.32 | 0.83 | 26.23 | 0.73 |
| GAN baseline | 20.02 | 0.81 | 24.79 | 0.88 | 20.32 | 0.48 | 18.06 | 0.34 | 26.77 | 0.71 | 23.92 | 0.59 |

### C.5    ADDITIONAL RESULTS OF VARYING z

We investigated the influence of the auxiliary variable $\mathbf{z}$ in Section 4.3. Here, we include more observations in Figure 9.

### C.6    ADDITIONAL QUALITATIVE RESULTS ON GENERATION

We present more generated image samples in Figure 14, 15, 17, and 18.

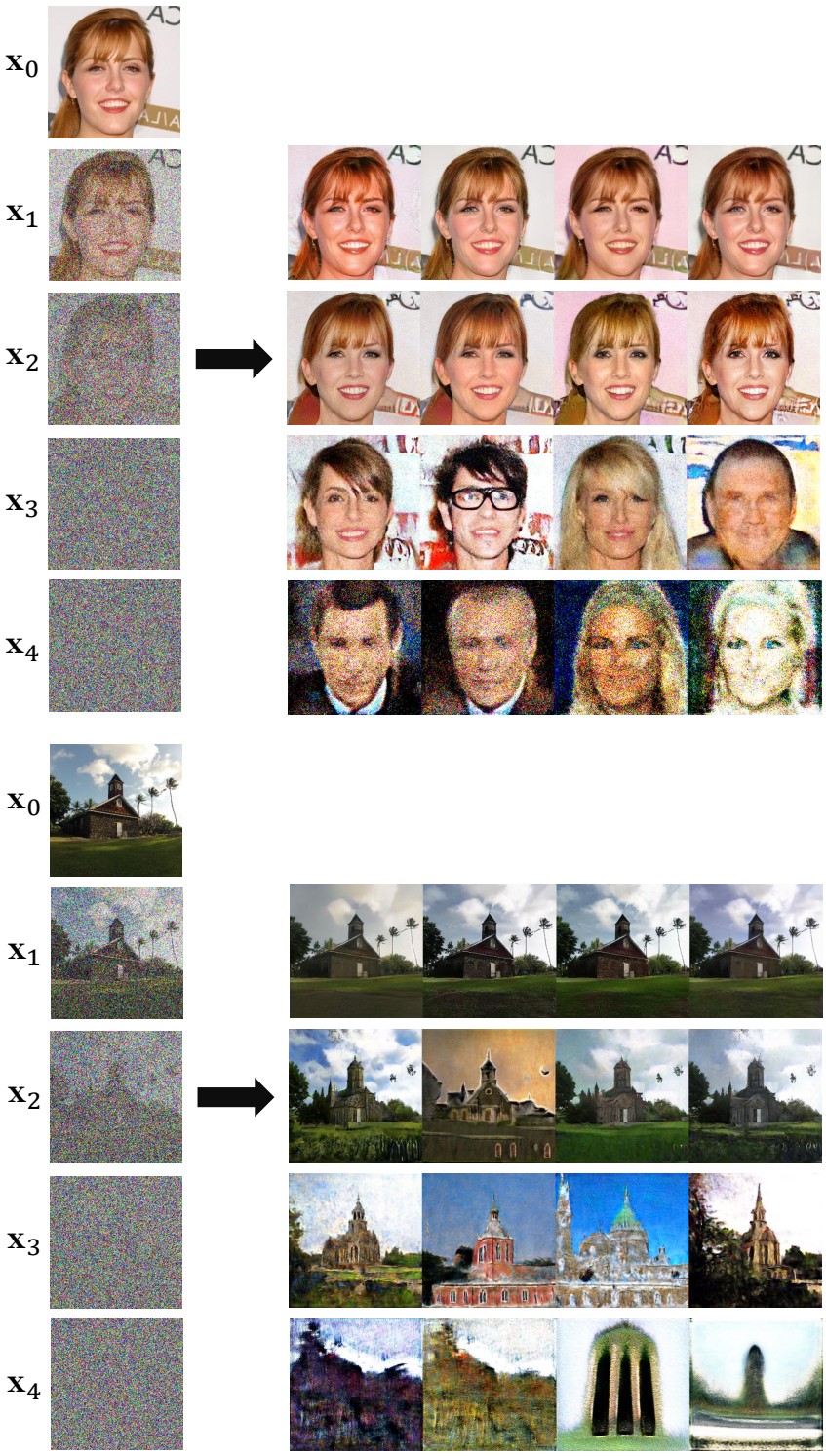

Figure 9: Illustration of the effect of varying $\mathbf{z}$ on CelebA-HQ (top) and LSUN (bottom). The images in the leftmost column depict the selected trajectory $\{\mathbf{x}_k\}_{k=1}^4$ degraded from an image $\mathbf{x}_0$. Each row on the right presents restored images of $\mathbf{x}_t$ using four different random auxiliary values $\mathbf{z}$. When the noise level is small, they generate almost identical images, which means that the restoration problem is almost well-posed. As the noise level increases, however, each degraded observation $\mathbf{x}_k$ estimates diverse images depending on the $\mathbf{z}$. In other words, the larger the noise, the more severe the ill-posedness, and the results validate that a much wider restoration is possible through the introduction of $\mathbf{z}$.

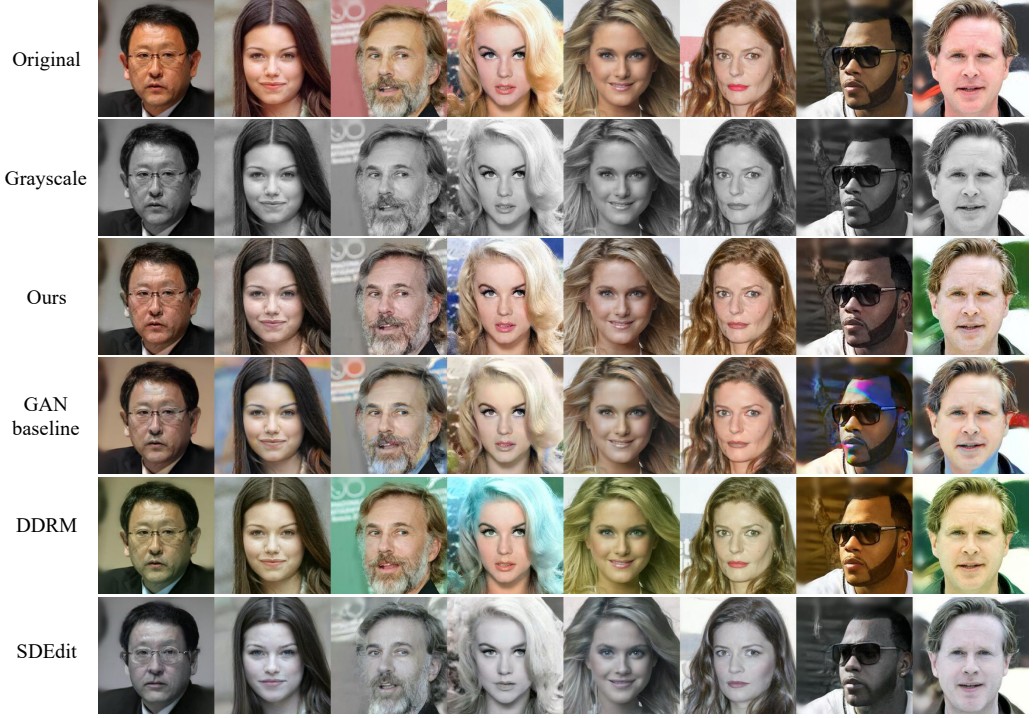

Figure 10: **Colorization**. Qualitative comparison on CelebA-HQ.

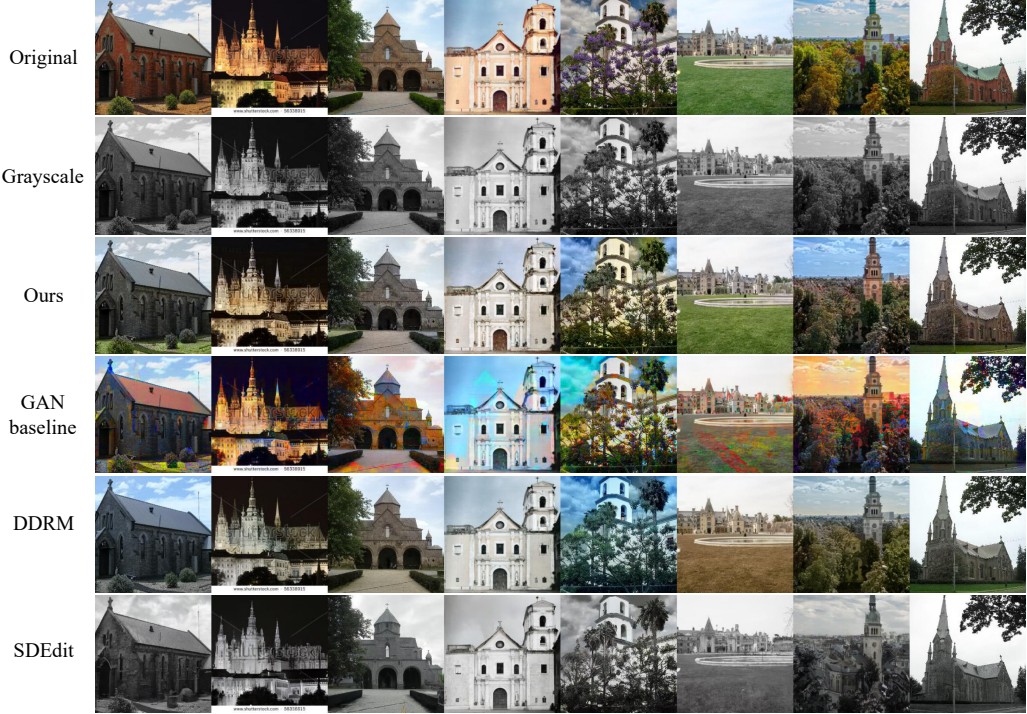

Figure 11: **Colorization**. Qualitative comparison on LSUN church.

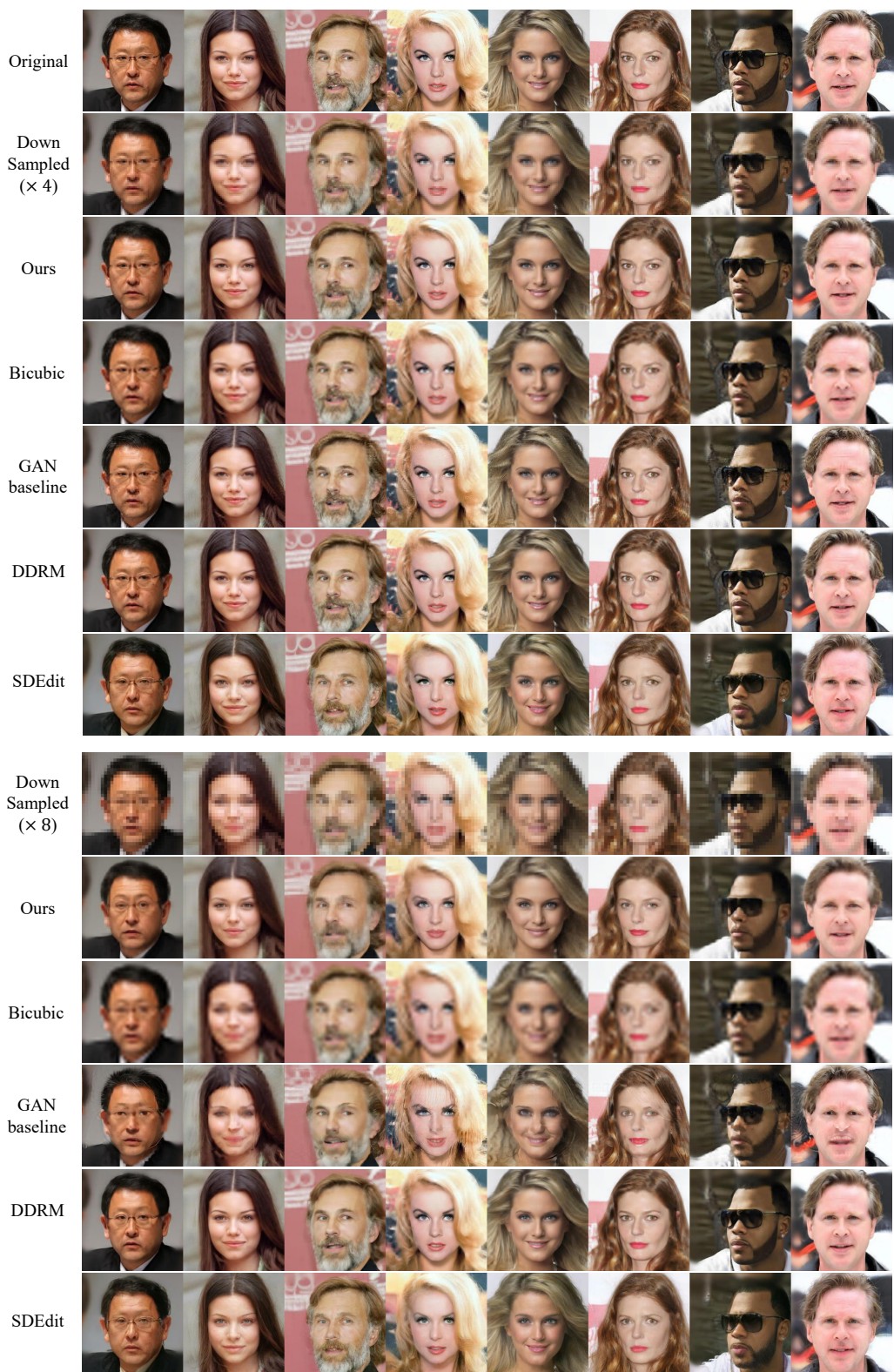

Figure 12: **Super-resolution**. Qualitative comparison on CelebA-HQ.

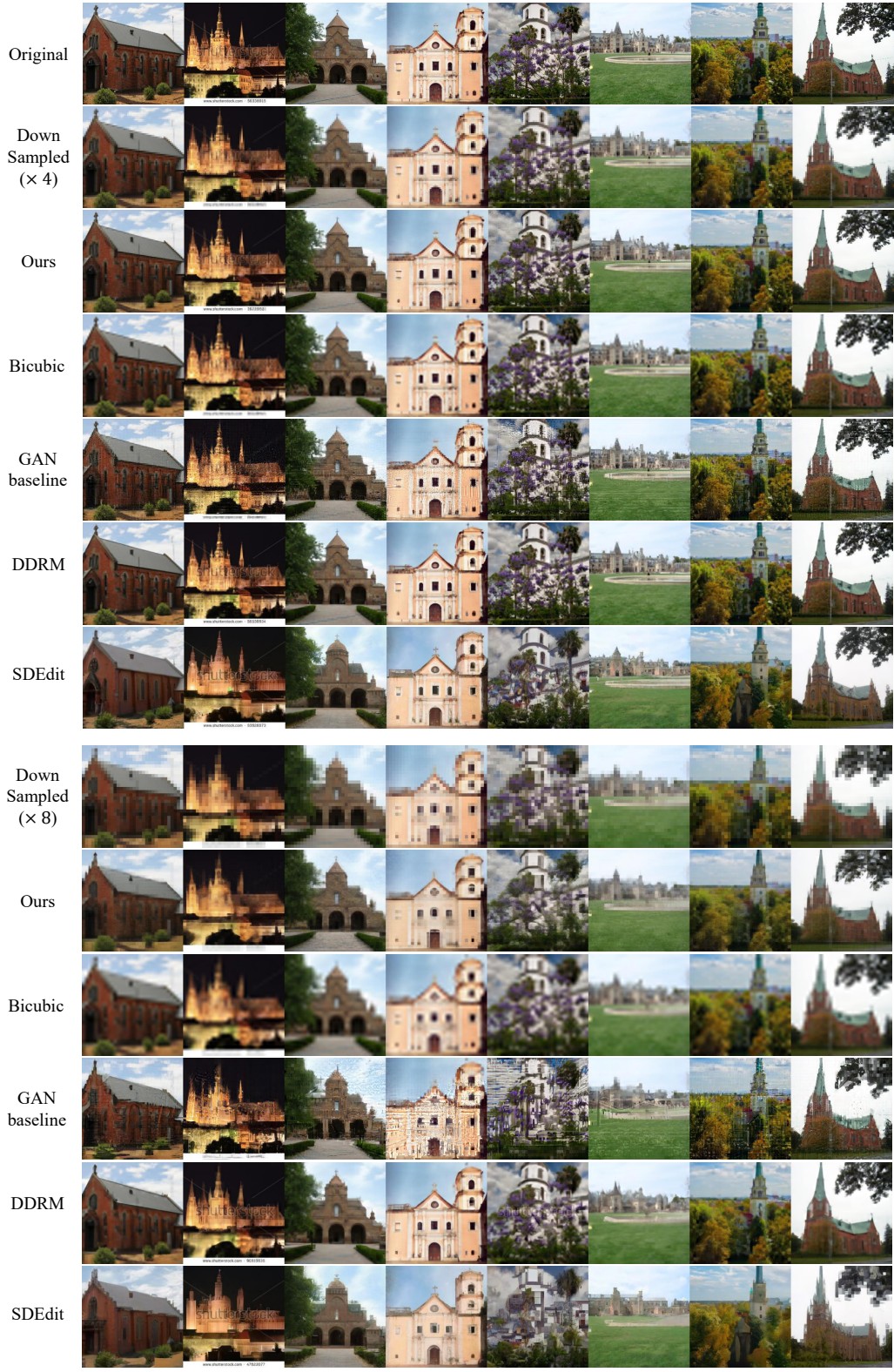

Figure 13: **Super-resolution**. Qualitative comparison on LSUN church.

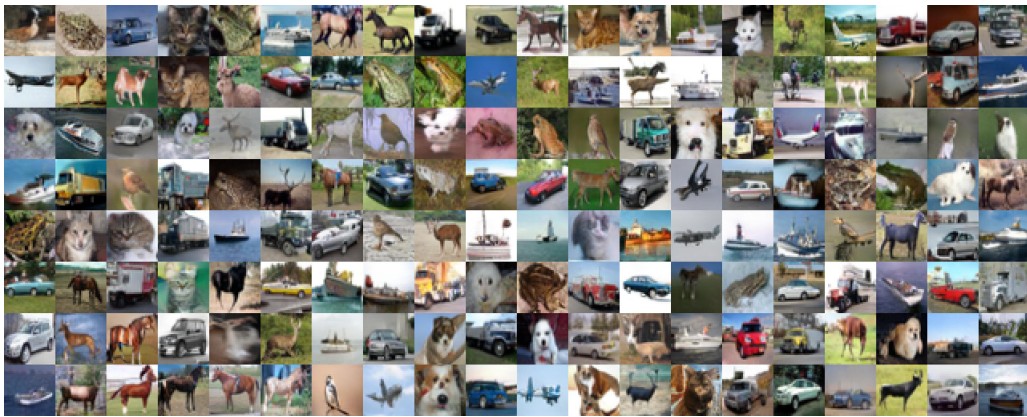

Figure 14: Generated samples of RGM-D on CIFAR10.

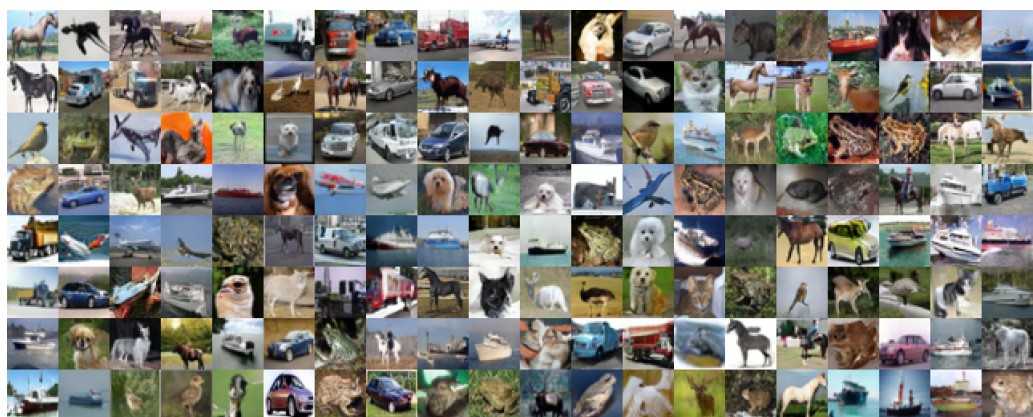

Figure 15: Generated samples of RGM-SR on CIFAR10.

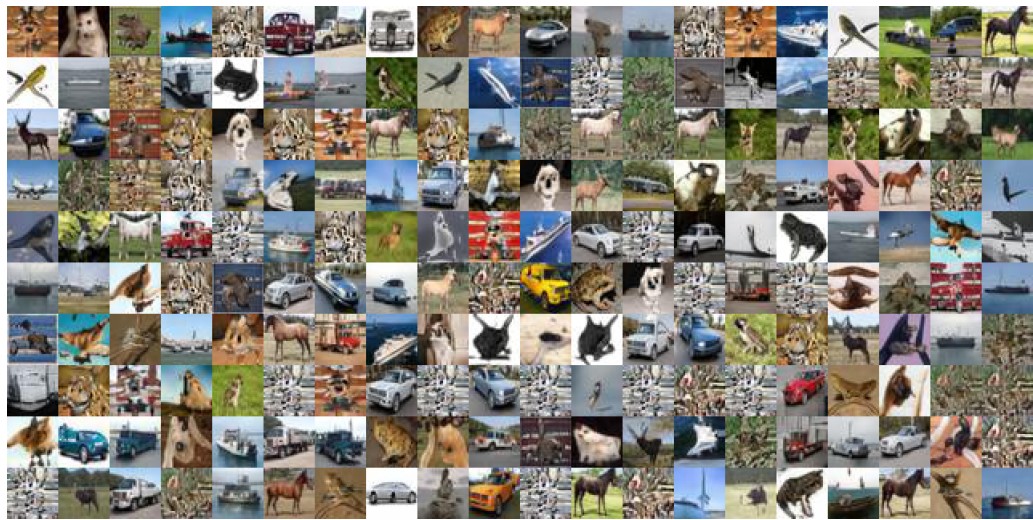

Figure 16: Mode collapse of RGM-D trained without the data fidelity term. Sampled images of RGM-D ($\lambda = \infty$) seem repetitive.

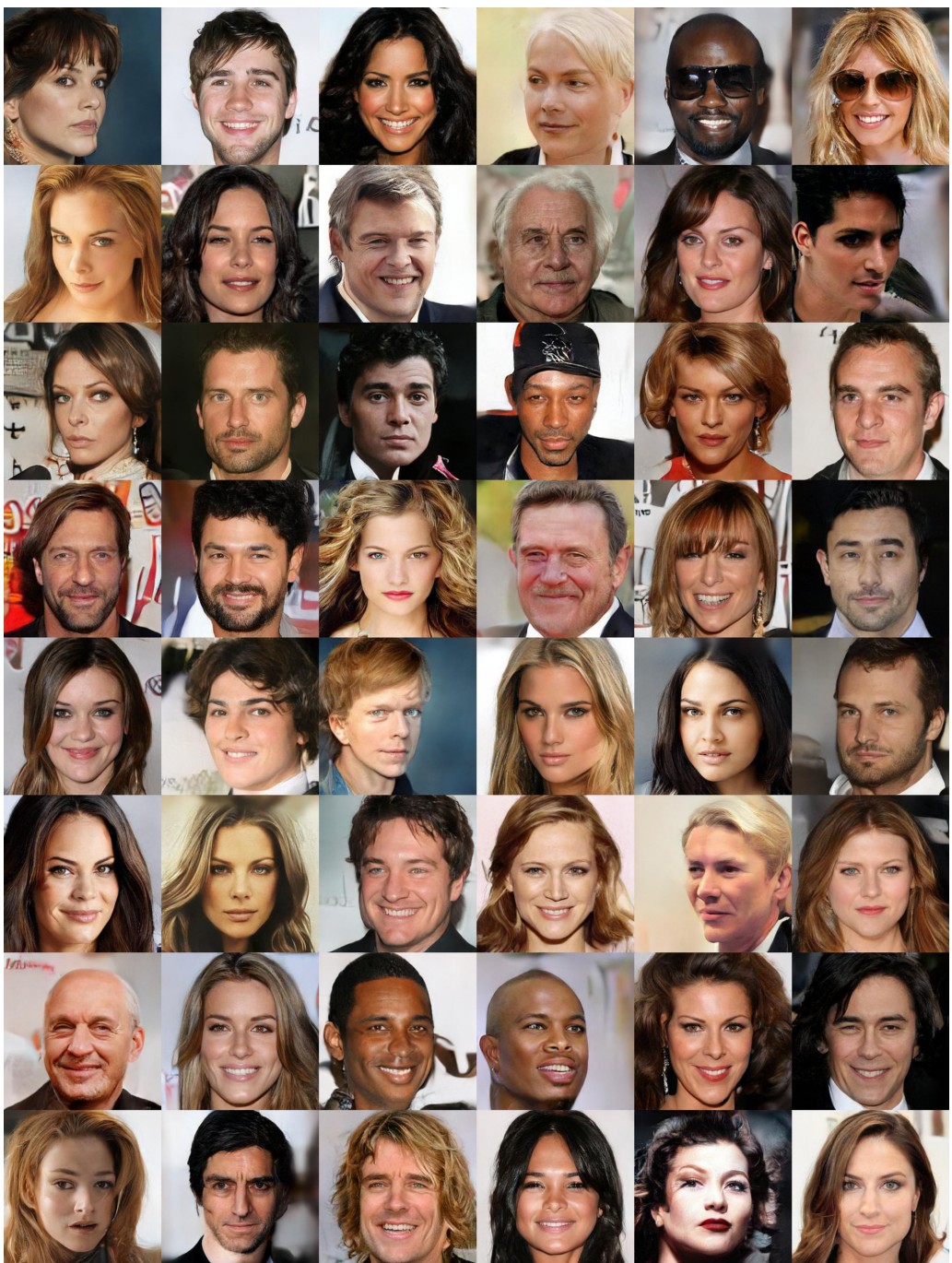

Figure 17: Additional qualitative results of RGM-D trained on CelebA-HQ-256.

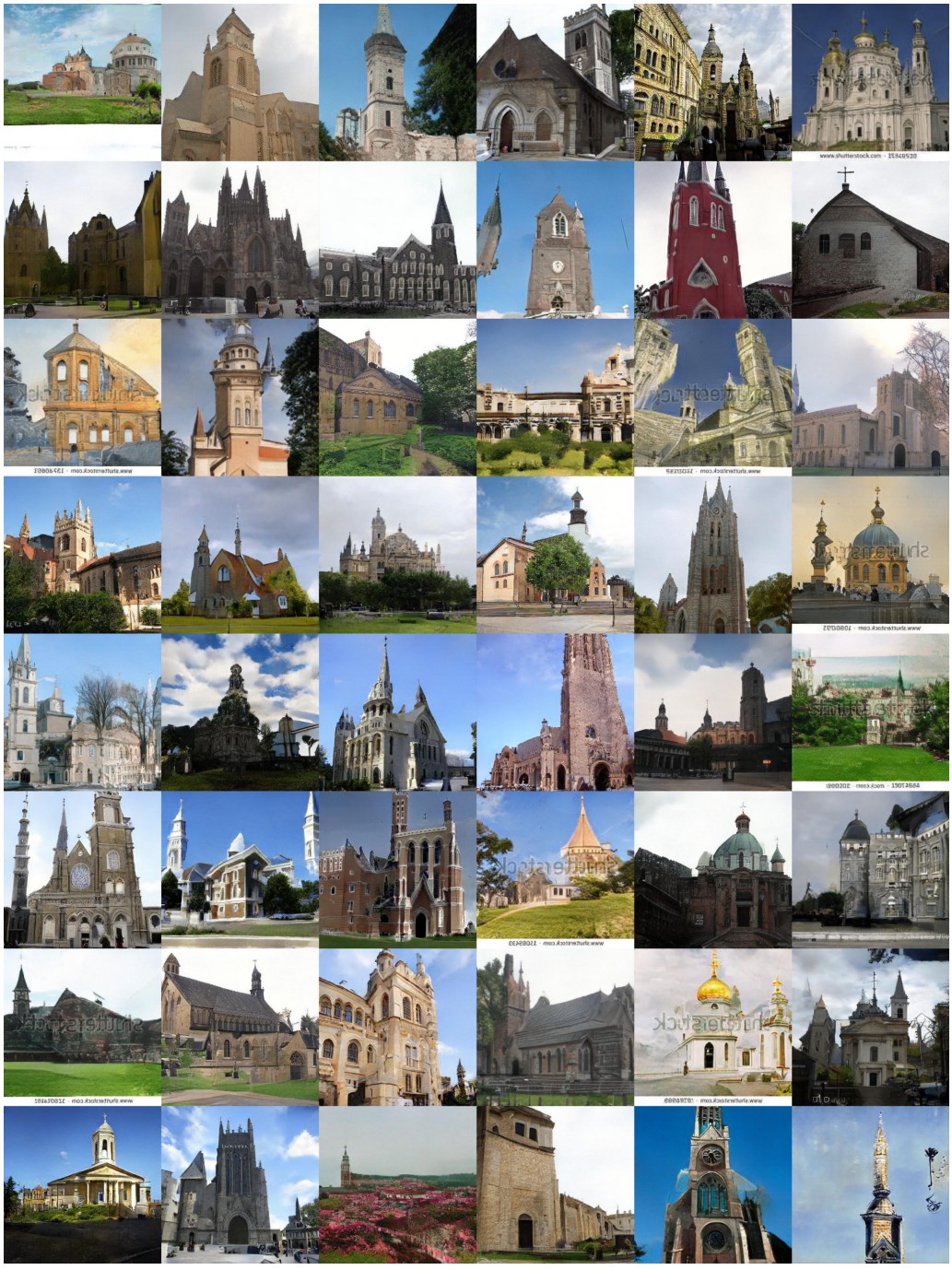

Figure 18: More qualitative results of RGM-D trained on LSUN Church.

