# OpenReview forum: "Restoration based Generative Models"
_ICLR.cc/2023/Conference — Submitted to ICLR 2023_

### Official Review · Reviewer_58kp · 2022-10-23

**Confidence:** 4
**Correctness:** 4
**Technical Novelty And Significance:** 4
**Empirical Novelty And Significance:** 4
**Recommendation:** 6

**Clarity, Quality, Novelty And Reproducibility:**

quality: This is a new attempt to use the generative model for graphics restoration, which is reasonable and effective. The proposed RGM uses MAP estimation instead of MMSE in DGMs, which greatly improves the efficiency of the model.

clarity: The overall writing and organization are good.

reproducibility: The authors provide code.

originality: The innovation of this paper has not appeared in other papers.

**Strength And Weaknesses:**

**strength**

-  The authors propose  a new flexible family of generative models, named restoration-based generative models (RGMs). RGMs achieve state-of-the-art performance using a limited number of forward steps.
- The proposed MAP estimation is efficacy, and a 2D example gives a visual validation.
- The ablation study results are extensive and demonstrate the effects of all parts of the MAP.
- The main paper and supplementary file are well prepared. The motivation is clear and reasonable. The paper is carefully organized.
- The authors extend RGM to general restoration and propose a new model established upon super-resolution (SR).
- The authors also provide code, which further shows the solidness of the work.

**weaknesses**

- The multi-scale training is introduced to alleviate the latent inefficiency of DGMs, where experiments are needed to prove it.
- Figure 4 is not mentioned in the article, and some introduction should be added.


**Summary Of The Paper:**

This paper proposes a generative model in terms of image restoration, named restoration-based generative model (RGM). This paper eliminates expensive sampling by performing MAP estimation and incorporating implicit prior information via GAN. Furthermore, a multi-scale training is proposed to alleviate the latent inefficiency of DGM. The ablation study demonstrates the effect of all parts of the MAP objective. A series of experiments on different datasets demonstrate that the RGM achieves state-of-the-art performance when the number of forward steps is limited.

**Summary Of The Review:**

This paper proposes a new generative model, named restoration-based generative models (RGMs). RGM uses MAP estimation and incorporates implicit prior information via GAN. Furthermore, RGM is not limited to denoising and can be effectively extended to other IR tasks, like image SR. The overall writing and organization are good. It is a novel and meaningful work.

---

> ### Author Response · Authors · 2022-11-08
> **Author response to reviewer 58kp**
>
> We thank the reviewer for valuable comments. We hereby carefully address the reviewer's comments as follows:
>
> We first answer the reviewer's comment on the latent efficiency of our multi-scale training of RGM-SR.
> The forward process of RGM-SR downsamples the image, in the case of CIFAR10, to a resolution of $4\times 4$. Therefore, the marginal distribution at the final forward step $T$ is low-dimensional. Recovering the image distribution from a low-dimensional distribution is termed as latent inefficiency in the original manuscript. The term latent efficiency seems to be a bit overused here, and we are sorry for the confusion caused here. We have clarified this in the revised version.
>
> We also appreciate for pointing out the missing cross-reference to Figure 4. We have updated Section 4 in the revised manuscript by addressing this.
>
> Finally, we would like to note that we added additional results in Appendix C.2 which demonstrate that our RGM framework universally works for variously parametrized prior terms, not being tied by the GAN structure introduced in Section 3.2. The proposed RGM is a flexible family of generative models that have degrees of freedom for the prior term and the forward process. And these bring us enhancement on efficiency and performance. We believe that our RGM framework paves the way for modeling flexible and efficient generative models.
>
> We hope that our response can address your main concerns. If so, we would like to kindly ask the reviewer to consider raising the score accordingly.

---

> > ### Comment · Reviewer_58kp · 2022-12-01
> > **Thanks for the feedback**
> >
> > Thanks for the feedback, which addressed my concerns. Given the other reviewers' comments and corresponding responses, I keep my rating score. Thanks.

---

### Official Review · Reviewer_3aDJ · 2022-10-24

**Confidence:** 3
**Correctness:** 3
**Technical Novelty And Significance:** 3
**Empirical Novelty And Significance:** 3
**Recommendation:** 6

**Clarity, Quality, Novelty And Reproducibility:**

The RGM is an interesting idea by imposing an additional generative model such as GAN as an implicit prior. The idea is new and the presented performance is good. However, due to the re-use of an additional generative model, some limitations are expected as described above.

Other points:

1. Can the authors explicitly show the sampling time of RGM compared with other generative models?

2. Will different parameters of GAN will affect the final performance of RGM? This is important since, if so, then it implies that RGM highly relies on the success of GAN.

**Strength And Weaknesses:**

Strengths:
1. A new perspective of RGM from IR is provides, which is interesting and opens up possibilities for improvements from denoiser design perspective.

2. Using MAP with a regularized prior imposed by GAN, RGM achieves very good performance on standard datasets, and in particular, with a very few  (tens of) samplings steps.

3. Extended applications in inverse problems are enabled.

Weaknesses:

1. The training process of RGM is much difficult than previous RGM, as shown in Algorithm 1.  In particular, a prior generator such as GAN is needed. In other words, two generative models are actually needed. Suppose that there is no GAN available at hand, then, to train a RGM, we have to first train a GAN, which itself is not an easy task. This kind of double-training is not preferred.

2. Also related to the use of GAN as implicit prior. Intuitively, it is unsurprising that the performance would improve if an additional generative model is imposed as a prior. The authors claim that the use of MMSE loss is not as good as MAP. The authors verified the benefits of MAP in Figure 3 in the toy model.  I am a bit suspicious of this point. How can we confirm that the improvement comes from the MAP, rather than the implicit prior imposed by the additional generative model GAN in RGM?

3. How is the performance of RGM for out-of-distribution datasets for inverse problems? It seems that by imposing the additional GAN prior, the obtained RGM is difficult to generalize to OOD datasets.


**Summary Of The Paper:**

This manuscript proposed a variant of generative models called restoration based generative models (RGM). The key idea is based on a new interpretation of denoting generative models (DGMs) from an image restoration (IR)  perspective. By replacing the MMSE denoiser with MAP denoiser and introducing a regularized prior imposed by another generative models such as GAN, RGM apparently reduces the number of sampling steps. RGM has also been applied to inverse problems.

**Summary Of The Review:**

Overall the idea of RGM is interesting. From the IR perspective, RGM is designed by  imposing an additional generative model such as GAN as an implicit prior. My  main concerns are about the essential need of an additional generative model such as GAN, which might affect the training, stability, flexibility, and efficiency of the proposed RGM as described above.

Update after rebuttal:

I raised the score accordingly.

---

> ### Author Response · Authors · 2022-11-08
> **Author response to reviewer 3aDJ**
>
> We thank the reviewer for valuable comments. We hereby carefully address your concerns as follows:
>
> **Q.1 In particular, a prior generator such as GAN is needed. In other words, two generative models are actually needed..**
>
> **Reply**:  We would like to note the reviwer that our model is one staged and end-to-end model, not a 2-staged model. (We do not train GAN for the first stage, and then train our model.) Our training algorithms learn the generator by the proposed MAP loss while learning the prior term via the discriminator.
> We would like to refer the reviewer to Algorithms 1 and 2 in Appendix B.2 for further details and clarity.
>
> **Q2. My main concerns are about the essential need of an additional generative model such as GAN..**
>
> **Reply**: First, we would like to emphasize that our RGM is neither tied nor relies on the GAN structure. Also, all of our improvement comes from adopting the MAP approach rather than from the help of GAN.
> Our RGM has the freedom to parametrize the prior term $g$ of MAP objective Eq.(7) without being limited by the GAN structure presented in Section 3.2. To demonstrate this, we design the prior term in two additional ways:
> Maximum mean discrepancy (MMD) [1] and distributed sliced Wasserstein distance (DSWD) [2].
> We have added the new experiments and discussions into the Appendix C.2, due to space limitations in the main paper.
> Figure 8 validates that RGMs with three different priors show consistent performance, where they are all more efficient than the MMSE estimator. Please note that MMD and DSWD do not learn priors. Despite this simple structure, Figure 8 confirms that our RGMs with MMD and DSWD are more efficient than MMSE approach.
>
> Furthermore, we run an experiment of RGM with the learnable DSWD prior term on CIFAR10. It achieves an FID score of 3.14, retaining comparable performance with RGM trained with the GAN prior. This verifies that our MAP estimation works universally well even when the prior term is parametrized in a way other than the GAN structure. We believe that our RGM framework paves the way for modeling flexible and efficient generative models.
>
> We hope that these new experimental results and our answer were able to address the reviewer's core point of concern that the performance and training of the RGM may depend on the GAN.
>
> **Q3. How is the performance of RGM for out-of-distribution datasets for inverse problems? ..**
>
> **Reply**: As the reviewer pointed out, our model cannot be generalized to out-of-distribution data for the restoration task. This is our limitation in solving inverse problems.
>
> **Q4. Can the authors explicitly show the sampling time of RGM compared with other generative models?**
>
> **Reply**:
> The sampling time of RGM-D is approximately 0.25 seconds to sample 100 samples on a single Tesla V100 GPU. According to [3], DDPM, Score SDE(VE) and FastDDPM take 80.5s, 421.5s, and 4.01s, respectively, to sample under the same conditions as ours (single Tesla V100 GPU, PyTorch implementation). we have added Appendix B.2 in the updated manuscript to address the training/inference time of our RGM.
>
> **References**
>
> [1] Dziugaite et al. 2016. Training generative neural networks via Maximum Mean Discrepancy optimization. Arxiv preprint.
>
> [2] Nguyen et al. 2021. Distributional Sliced-Wasserstein and Applications to
> Generative Modeling. ICLR.
>
> [3] Xiao et al. 2021. Tackling the Generative Learning Trilemma with Denoising Diffusion GANs. ICLR.

---

> > ### Comment · Reviewer_3aDJ · 2022-12-01
> > **Thank the authors' feedback**
> >
> > Sorry for my late reply. Thank the authors for clarifications of some of my misunderstandings and their effort in improving the quality.   I raised my score accordingly.

---

### Official Review · Reviewer_otH9 · 2022-10-25

**Confidence:** 4
**Correctness:** 3
**Technical Novelty And Significance:** 3
**Empirical Novelty And Significance:** 3
**Recommendation:** 5

**Clarity, Quality, Novelty And Reproducibility:**

The paper's writing could be greatly improved.
First, the acronym DGM is typically used for Deep Generative Models, not for Denoising-based Generative Models. Please use diffusion models instead.

There are several typos, spelling and grammatical mistakes. Some statements in the paper are also not true. Below there are some examples:

* In the Introduction, it is not true that in diffusion models the latent and the data always possess the same dimension. Latent Diffusion Models (e.g. Stable Diffusion) diffuse in a latent space (encodings of natural images).
* Eq. (4) is only true for specific types of samplers. E.g. in DDIM, the sampling process is not Markovian.
* what do the authors mean by "immovable diffusion"?
* what do the authors mean by "latent inefficacy"?
* a multiscale training that resolve => [...] that resolves
* denoising. deblurring, super-resolution and inpainting are all different inverse problems (Section 2). Also, in the Introduction, Image Restoration is not an inverse problem. It is a term that describes a family of inverse problems.
* The acronyms VPSDE, VESDE are not explained (or even cited).
* In Equation (5), the neural network should also take as input the timestep t.
* In Section 3.1, the authors are effectively repeating the loss of Eq. (5) while presenting it as something that is changing.
* It would be preferable to use more neutral/academic writing in some places. E.g. "set their mind on" -> consider. Also, "tremendous amount of solutions": the amount of solutions is either finite or infinite.
* "whereas DGMs use T=1000, 2000 steps". Not true for many models/samplers.
* "a latent as much as dimension of pixel space": There is a grammatical mistake here. Not sure what the authors are trying to say.
* the links in the paper are not clickable.


**Strength And Weaknesses:**

Strengths:
* The ideas presented in the paper are interesting. It is intuitive that better priors, introduced by a different forward model and a regularization term, can lead to better models.
* Experimental results are quite convincing.
* The observation that optimizing for super-resolution leads to better results for super-resolution is interesting. There are many recent papers that generalize diffusion to non-gaussian corruptions (e.g. Cold Diffusion, Soft Diffusion, Blurring Diffusion Models, etc) and it would be interesting to evaluate whether these models also perform better on the inverse problem they train for.


Weaknesses:
* The paper is poorly written. See analytical comments below.
* The classical objective *is* a MAP method. As long as there is no stochasticity in the solution, you are maximizing something -- the only thing that is changing is your prior (imposed by the regularization term).
* It is interesting that you introduce stochasticity to account for the ill-posedness nature of the inverse problems. However, what prevents the model from dismissing entirely the randomness? It is just an additional input, right? The model can always ignore it. Why is it best to not ignore it for minimizing the objective?
* I am struggling to understand the added regularization. It seems that it is a trainable Discriminator (similar to the Discriminator in GANs). If that's the case, then you should also feed to the discriminator the real images, right? Eq. (8) doesn't make sense to me -- if the Discriminator doesn't see real images, how is it regularizing? How is the generator not fooling the discriminator? Now, even if we assume that there is a typo and the Discriminator actually sees real images, doesn't that introduce the training instabilities of GANs? Or mode collapse and less diversity? Please clarify these points in the rebuttal.
* The theoretical underpinnings of the proposed objective are not discussed/explained. When there is no regularization term, it has been shown that the DSM objective learns the score function (and minimizes the KL between the true and the sampling process). Can similar guarantees be achieved for this new objective?
* How is the sampling rule derived by using this objective? Since there is no guarantee that you are learning the score-function, can you still solve for the reverse SDE and get reasonable samples? Can we compute likelihoods? If so, how? Would be interesting to add likelihood evaluations to the paper.
* The paper has several inaccurate and/or unsupported claims. It is not true that all diffusion models diffuse in the pixel space, e.g. see Latent Diffusion Models. It is also not true that Diffusion models typically use thousands of steps. The is a huge line of work for accelerating diffusion models, including Progressive Distillation. The state-of-the-art claim for CelebA-HQ-256 is not exactly accurate. From a quick search, it seems that StyleSwin (which is used as a baseline for the inverse problems) achieves FID 3.25, which is 2x better.
* From the experimental evaluation of the paper, it seems that the main contribution to the improvement in the FID scores is the change of the forward process. RGM-D performs *worse* than the simple VE SDE. There are several recent works that use different distributions for corrupting images, e.g. Cold Diffusion, Soft Diffusion, Blurring Diffusion Models, Generative Modelling With Inverse Heat Dissipation, etc. Since the improvement is coming from the change of corruption, these other works should be discussed more extensively. Ideally, a performance comparison with these works should be made.


**Summary Of The Paper:**

The paper proposes three changes to vanilla diffusion models: i) a regularization term in the training objective, ii) a source of randomness in the training objective to account for the fact that there is no single solution to the denoising problem iii) a class of more general corruption processes. The paper demonstrates that the resulting models are faster to sample from and they achieve better or on par with the baselines.


**Summary Of The Review:**

The paper presents some interesting ideas: GAN regularization in the training objective, different forward processes and a stochastic knob in the input of the generative model. However, there are several weaknesses (detailed above) that prevent me from recommending acceptance. The most important issue for me right now is that the main benefits seem to be coming from the change of the forward process and not from the regularization. If that's the case, I think the presentation of the paper needs a lot of restructuring and the baselines for the comparisons should be other methods that generalize diffusion.

Update, Nov. 13: The authors addressed some of my concerns in the rebuttal and I am increasing the score from 3 to 5. I believe that there are still some concerns about the novelty of this work, but I am open to more discussion and perhaps a further increase of the score if these issues are addressed.

---

> ### Author Response · Authors · 2022-11-08
> **Author response to reviewer otH9 (2)**
>
> **Q5. How is the sampling rule? Can you still solve for the reverse SDE?**
>
> **Reply**: The sampling procedure has been summarized in Algorithm 3 in Appendix B.2. Since denoising diffusion models (DDMs) are based on MMSE, it is difficult to recover the data distribution from highly degraded data. (See Figure 3.) Therefore, DDMs divide between the Gaussian latent and the data distribution into small pieces, and the corresponding forward process is described by a linear SDE as $T\rightarrow \infty$. On the other hand, our RGMs, which are based on MAP, can approximate the data distribution much more efficiently due to the existence of the prior term. Therefore, RGMs use much less $T$ (e.g. $T=4$ for RGM-D) and learn a restorator $G_\theta$ instead of learning the score function for SDEs. In other words, DDMs generate images through the reverse SDE because MMSE-based approach requires large forward steps to obtain high-quality data. In contrast, since our RGMs can produce high-quality data using much fewer steps, sampling is performed as provided in Algorithm 3 rather than reverse SDE.
>
> **Q6. The paper has several inaccurate..**
>
> **Reply**: We appreciate the reviewer for pointing this out.
> We have revised the manuscript by addressing the reviewer's comments.
> We also added the FID score of Progressive Distillation in Table 1 (CIFAR10) and the FID of StyleSwin in Table 2 (CelebA-256).
>
> **Q7. From the experimental evaluation..**
>
> **Reply**: We agree with the reviewer that it is instructive to compare with the related works suggested by the reviewer. We have added a comparison to these models in Appendix C.3. In comparison with our model itself, as the reviewer pointed out, the change in the forward process brings FID improvement.  But compared to other models, Table 8 shows how efficiently our MAP-based approach produces high-quality images. Our RGMs achieve superior performance with very few NFEs. The choice of the forward process would significantly affect the performance of the model, as it determines how to bridge the data and the latent distribution. However, rather than changing the forward degradation process, how to recover the data distribution from this degradation seems to be more crucial. And the Table 8 shows that our RGMs offer an effective way of generative modeling from image degradations.
>
> **Q8. Typos and grammatical mistakes.**
>
> **Reply**:  We are grateful for pointing out the typos and grammatical mistakes that we missed. We have included them in the updated manuscript. (The full manuscript together with appendices is provided in the supplementary material.) We have also replaced acronyms of denoising diffusion generative models from DGMs to DDMs.
>
> We hope that our response can address your main concerns. If so, we would deeply appreciate it if the reviewer reconsider the novelty and contribution of our paper.

---

> > ### Comment · Reviewer_otH9 · 2022-11-14
> > **Follow-up on rebuttal**
> >
> > I want to thank the authors for their efforts to improve the paper during the rebuttal period. I increased my score from 3 to 5.
> > I still have some concerns, that I detail below.
> >
> > * The training procedure seems clear now, I want to thank the authors for clarifying. However, some of the related questions I had are not discussed in the rebuttal. Specifically, adversarial training seems to be coming with its own disadvantages; mode-collapse, training instabilities, etc. Is there any reason why this framework won't experience such effects?
> > * The authors clarify in the rebuttal that FID benefits over prior methods are mostly coming from the change of the corruption process. The authors claim that the (adversarial) regularization term is useful because it decreases the number of sampling steps needed. This is a fair statement. However, this has already been observed in the prior work of Xiao et. al, where adversarial losses are also used. The authors acknowledge this in the Appendix and they claim that their method outperforms prior work in terms of training speed (less iterations are required). This is a valid contribution over prior work and is evaluated as such. I would highly recommend clarifying the connection to prior work further in the main body of the paper.
> > * Following up on my comment regarding the theoretical understanding: I still feel this is a limitation of the paper. For example, by switching to this new loss, the connection to score-matching is lost and there is no obvious way of: i) applying tricks that have been widely successful in the literature, such as classifier-free guidance, ii) computing likelihoods, iii) finding deterministic samplers. Also, I still don't understand the theoretical interpretation of this work. What does it mean "near optimal" in the sentence just above Eq. 9?
> >
> >
> > Minor:
> > * It seems that the authors used some tool to highlight (with red color) the differences between the previous and the current version. The tool introduced some artifacts in the typesetting of the document, especially the mathematical equations. I encourage the authors to fix this issue.
> > * It seems that the authors cite twice the paper: "Tackling the generative learning trilemma" with denoising diffusion gans".
> > * The generator in all the equations should be taking also the time as an input.
> >
> > Side note: I think it would be interesting to evaluate further the effect of randomness as an extra input. It seems to be giving a big boost in performance and I am wondering whether it would be a trick that could find broader applicability in the diffusion literate. Giving the short remaining rebuttal period, I am not asking the authors for an ablation on that - just wanted to point out that I find this interesting and unexpected.

---

> > > ### Author Response · Authors · 2022-11-14
> > > **Thank you for feedback**
> > >
> > > We warmly thank the reviewer's further valuable suggestion and comments. Please find our responses below.
> > >
> > > **Q1. Disadvantages from adversarial training.**
> > >
> > > **Reply**: As demonstrated in Table 3, we found that separating the generation process into several steps ($T>1$) leads to significant improvement in the FID score by reducing the load of the model. Moreover, the data fidelity term resolves the mode collapse, which have already been discussed in Section 4.3 and Figure 15. We believe that the model training has been stabilized thanks to these.
> > >
> > > Furthermore, we would like to introduce additional results added in Appendix C.2 to the revised manuscript (to address the other reviewer's comments), which demonstrate that our RGM framework universally works for variously parameterized prior terms, not being tied by the adversarial training of GAN introduced in Section 3.2.
> > > In these new experiments, we design the prior term in two additional ways: Maximum mean discrepancy [1] and distributed sliced Wasserstein distance (DSWD) [2].
> > > Results in Figure 3 and the FID score of RGM with DSWD prior on CIFAR10 confirm that our RGM framework achieves consistent performance (more efficient than MMSE) no matter how the prior term is parameterized.
> > > These results validate that our RGM is not tied to or dependent on the GAN structure.
> > >
> > > **Q2. Clarifying the connection to the prior work of Xiao et. al.**
> > >
> > > **Reply**: Following the reviewer's suggestion, we have addressed the connection and comparison with DDGAN (Xiao et. al.) in Section 5.
> > >     DDGAN enhanced sampling speed through the usage of GAN, which may look similar to our model. However, as mentioned above, we show through the additional experiments (in Appendix C.2) that, unlike DDGAN, our RGM framework does not rely on GAN structure because we leverage the MAP approach that DDGAN did not suggest. These conforms that our performance and sampling acceleration are not due to GAN, but because we adopted the MAP approach. This is another difference with DDGAN.
> > >     Also, as the reviewer mentioned, our RGM requires less training iteration than DDGAN to achieve the same performance.
> > >
> > > **Q3. Theoretical understanding and connection to score-matching.**
> > >
> > > **Reply**: Our work has begun with the observation that the objective of the existing denoising-based generative models is based on MMSE approach and with the purpose of resolving the sampling inefficiency resulting from the MMSE through the MAP approach. The story of score matching loss is possible when the sampling step of the MMSE approach is sent to infinity, that is, when the discrete forward process is described as an SDE (a continuous time dynamics). In other words, the purpose is quite different from ours.
> > > Our MAP objective considerably improves the sampling efficiency, but it is true that continuous time sense of score matching was lost. Therefore, properties that are derived from SDE, such as deterministic sampling, cannot be achieved in our framework. However, please note that instead of losing these, we gained lots of advantages: enhancing the sampling efficiency as well as improving the performance by changing the corruption process. We further showed the applicability of our RGMs to imaging inverse problems and we believe this will lead to many potential applications.
> > >
> > > We have also clarified the words 'near optimal' in the sentence just above Eq. (9) in the revised manuscript.
> > >
> > > Furthermore, the analysis of the effect of randomness pointed out by the reviewer seems to be a very interesting research topic. We thank the reviewer for pointing out such an worthwhile direction for future research and we believe that it could be studied more rigorously in future research. We have added this as future work in Section 6.
> > >
> > > **Q4. Minors.**
> > >
> > > **Reply**: We appreciate the reviewer for pointing out some mistakes that we missed.
> > > Revisions have been made in the updated manuscript to address all of these issues.
> > >
> > > Again, we'd like to sincerely thank the reviewer for your time and thoughtful feedback.
> > > We hope we had fully addressed the reviewer's comments and concerns. We do hope these responses and the revised manuscript are helpful for the reviewer to re-evaluate our paper.
> > >
> > > **References**
> > >
> > > [1] Dziugaite et al. 2016. Training generative neural networks via Maximum Mean Discrepancy optimization. Arxiv preprint.
> > >
> > > [2] Nguyen et al. 2021. Distributional Sliced-Wasserstein and Applications to Generative Modeling. ICLR.

---

> ### Author Response · Authors · 2022-11-08
> **Author response to reviewer otH9 (1)**
>
> We thank the reviewer for your comments and suggestions. We hereby carefully address your concerns as follows:
>
> **Q1. Main benefits seem to be coming from the change of the forward process and not from the regularization.**
>
> **Reply**: It seems the reviewer has a bit of confusion, which led to doubt about our contributions. So, we would like to first answer the reviewer's main concern on whether the main improvement of our model comes from the prior term or the change of the forward process. Our answer is **both**.
> Our model has two main parts:
> 1. Our MAP approach greatly improves the sampling efficiency. We achieve FID score 3.05 on CIFAR 10 with only four forward steps. Also, the results depicted in Figure 3 validate that our MAP-based approach is much more efficient than MMSE approach.
> The reviewer seems to have regarded that our MAP objective just adds a regularization term to the MMSE, which is the objective of diffusion models, but this is not.
> MMSE and the data fidelity term of our MAP objective are different. Also, due to the role of the prior term, which the reviewer refers to the regularization term, it is possible to generate high-quality images with a few sampling steps.
>
> 2. FID scores are further improved by changing the forward process (FID 2.47 on CIFAR 10).  As the forward process affects the performance of the model a lot, we have conducted ablation studies on this in Section 4.3. In addition, a comparison with existing models using various forward processes has been also added in the Appendix C.3 in the revised manuscript following the reviewer's suggestion.
>
> In other words, the MAP approach improves the sampling efficiency and changing the forward processs further enhances the performance of our model.
>
> **Q2. However, what prevents the model from dismissing entirely the randomness? It is just an additional input..**
>
> **Reply**: As the reviewer stated, we used additional randomness to alleviate ill-posedness of inverse problems. Imposing randomness into the model input might give additional stochasticity to the sample and thus relieve the ill-posedness. We empirically verified its effectiveness in the ablation study (Table 3). This randomness improved FID by more than 0.8.
>
> **Q3. I am struggling to understand the added regularization..**
>
> **Reply**: Eq. (8) is the loss for our generator $G_\theta$. When updating the discriminator, we put information from real images. We refer the reviewer to Algorithm 1 in Appendix B.2 for more clarification. We have clarified this in Section 3.2 in the revised manuscript.
>
> **Q4. The theoretical underpinnings of the proposed objective are not discussed/explained..**
>
> **Reply**: In Section 3.2, we have discussed that our objective can be regarded as minimizing Eq. (9).

---

### Official Review · Reviewer_VqGo · 2022-10-26

**Confidence:** 4
**Correctness:** 4
**Technical Novelty And Significance:** 3
**Empirical Novelty And Significance:** 2
**Recommendation:** 6

**Clarity, Quality, Novelty And Reproducibility:**

Clarity: Reasonably clear. In addition to the points listed above, the multi-scale RGM could be much better explained in the main paper though (based on the appendix, I believe it adds Gaussian noise as well as downsampling, which is not clear from the main paper).

Quality: see above.

Novelty: Novel AFAIK.

Reproducibility: Good. Code is provided.

**Strength And Weaknesses:**

Strengths:
- The method is an interesting combination of GANs with diffusion modelling which shines some light on the benefits of each.
- Impressive empirical results, including state-of-the-art on CelebA-HQ at 256 resolution.

Weaknesses:
- The output of $G_\theta(y,z)$ is repeatedly referred to as a "MAP estimate". This is not true; it is the stochastic output of a conditional GAN. Calling it a MAP estimate is misleading and should be changed.
- I cannot find the training times listed anywhere, only the number of training iterations. I imagine the proposed method takes more time per iteration than a diffusion model due to the more complex GAN objective, and this should be reported. Similarly, it's not clear to me if each function evaluation at test-time is slower than for typical diffusion models. A statement on this would be interesting.
- Progressive distillation ([Salimans & Ho, 2022](https://arxiv.org/abs/2202.00512), [Meng et al., 2022](https://arxiv.org/abs/2210.03142)) would be relevant work to comment on, since it speeds up sampling from diffusion models. [Salimans & Ho, 2022](https://arxiv.org/abs/2202.00512) achieve a FID of 3.0 on CIFAR10 with 4 NFEs and should probably be added to Table 1.

Minor:
- ~~It is not clear what metric is being reported in Table 4.~~

**Summary Of The Paper:**

This paper proposes a method to map from Gaussian noise to images by successively "removing'' a series of degradations. In one proposed variation, these degradations are the addition of Gaussian noise, as in the diffusion model framework, and in another variation they include downsampling. The authors train a conditional GAN to stochastically "undo" each degradation.

**Summary Of The Review:**

This paper presents shows that a combination of diffusion models with the GAN literature can obtain competitive results, which I believe is a good contribution. The weaknesses I listed were generally related to presentation and I am recommending acceptance as long as this is improved.

---

> ### Author Response · Authors · 2022-11-08
> **Author response to reviewer VqGo**
>
> We thank the reviewer for the thoughtful feedback. We hereby carefully address your concerns as follows:
>
> We first thanks for pointing out that $G_\theta(\mathbf{x},\mathbf{z})$ is incorrectly referred to as MAP estimator. We definitely agree with this. We have fixed this in the revised manuscript.
>
> We also appreciate the pointers to the references. We have included them in our related work (Section 5 of the paper). We also added the FID score of [1] in Table 1. And we have clarified the forward process of the multi-scale RGM in Section 4 (a paragraph named "setup") in the revised version.
>
> Lastly, we have added Appendix B.2 in the updated manuscript to address the training/inference time.
> We use the same architecture as NCSN++, which is used in ScoreSDE (VESDE) [2], except for the latent embedding term (embedding $z$). We found that the sampling time of RGM-D is approximately 0.25 seconds, which is more than 1000 times faster than ScoreSDE. Moreover, the training time of RGM-D on the CIFAR10 dataset is about 40 hours on 4 Tesla V100, PyTorch 1.10 implementation, which is smaller than ScoreSDE (more than 70 hours or more on PyTorch implementation).
>
> Hopefully, our replies have addressed all the concerns. If our replies feel satisfactory, we would like to kindly ask the reviewer to consider raising the score accordingly.
>
> **References**
>
> [1] Salimans & Ho. 2022, Progressive distillation for fast sampling of diffusion models. ICLR.
>
> [2] Song et al. 2021. Score-based generative modeling through stochastic differential equations. ICLR.

---

> > ### Comment · Reviewer_VqGo · 2022-11-17
> > **Thank you for the response**
> >
> > Thank you for making these changes. It remains unclear to me though why any of the method should be described as MAP estimation (as in e.g. the title of Section 3.2), since the GAN objective is not designed to yield a MAP estimate. I also don't think that a MAP estimate would even be desirable (since MAP estimates of distributions in high-dimensions are not necessarily representative of samples from those distributions).

---

> > > ### Author Response · Authors · 2022-11-18
> > > **Thank you for the feedback!**
> > >
> > > We would like to thank the reviewer for the comments and address the concerns in the following.
> > >
> > >
> > > As the reviewer pointed out, our objective is not a statistically complete MAP estimation since we do not exactly learn a log-likelihood of the data distribution as a prior. However, the image restoration literature often discusses MAP estimator at a higher level.
> > > Since the data density is not analytically tractable in practice, they replace this with a term that can give information about the prior density and also refer to the approximated objective as a MAP objective (We also provide several references [1,2,3,4]).
> > > Moreover, we have shown that our objective approximately learns an entropy-regularized MAP objective in Eq. (9).
> > > But, by addressing the reviewer's concern, we change the term 'MAP estimation' used to refer to our model to 'MAP-based estimation' in the revised manuscript.
> > >
> > > We also agree with the reviewer's comment that representing the distribution by recovering the density around a point may not be desirable in high dimensions. We believe that this worrisome is alleviated through two parts of our methodology: (i) separation of the generation process into several steps ($T>1$) and (ii) introduction of the auxiliary random variable $z$. The effect of both has been analyzed through the ablation study in Section 4.3. The results reported in Table 3 show that our model with $T=1$, which recovers the data distribution through direct our MAP-based estimator, has difficulties in learning the distribution, falling short of the FID score by 14.6.
> > > Compared to our model with $T=4$, which achieves the FID score of 3.17, it can be confirmed that the multi-step generation brings a great effect.
> > > Table 1 also shows that the auxiliary variable provides further performance improvements.
> > > Moreover, Table 2 demonstrates the competitive performance of our MAP-based model in high-dimensional data, obtaining the state-of-the-art FID score among restoration-based generative models.
> > >
> > > Finally, we would like to note that our MAP-based framework brings us significant enhancement in efficiency and achieves the comparable or better  performance than existing restoration-based generative models (See Tables 1 and 8).
> > >
> > > Again, we sincerely thank the reviewer for thoughtful feedback.
> > > We hope that our response can address all of the reviewer's concerns.
> > > Thank you very much.
> > >
> > > **References**
> > >
> > > [1] Ren et al. 2019, Simultaneous fidelity and regularization learning for image restoration. IEEE transactions.
> > >
> > > [2] Chen et al. 2019, Unsupervised Lesion Detection via Image Restoration with a Normative Prior. International Conference on Medical Imaging with Deep Learning
> > >
> > > [3] Yeh et al. 2019, Image Restoration with Deep Generative Models. IEEE.
> > >
> > > [4] Ren et al, 2020, Neural Blind Deconvolution Using Deep Priors. CVPR.

---

### Author Response · Authors · 2022-11-08
**Author response to all reviewers**

First of all, we deeply thank all the reviewers for spending their time carefully read our manuscript and their thoughtful feedback. We think that the reviewers raised several insightful questions and we believe that answering those questions has significantly improved our work.  We read the reviewers' comments carefully and have updated the paper to incorporate reviewers' advice and further improve the writing.
Below, we address the individual comments to each reviewer.
We hope that our response could address all questions and concerns. We deeply thank again all the reviewers for their kind and valuable comments.

---

### Author Response · Authors · 2022-11-28
**Reminder**

Dear reviewers,

we would like to kindly remind you that the period of discussion stage 1 is over,
and the period of the stage 2 discussion ends in two weeks.
We would appreciate it if the reviewers could take a look at our responses, and please let us know if our replies address all your questions/concerns
or if you have further follow-up questions.

We appreciate all reviewers for their valuable comments.
Also, we deeply thank the reviewers for their devoted time, energy, and effort.

Thank you very much.


$\ $

Best regards,

the authors of the paper

---

### Decision · Program_Chairs · 2023-01-20

**Decision:**

Reject

**Justification For Why Not Higher Score:**

The interpretation of the loss as a MAP estimate still seems incorrect. The GAN-based prior seems more of a regularization term than a "learned prior": while at convergence GANs minimize JS-divergence, in practice one does not train a discriminator to optimality. Thus, the GAN term acts more like a regularization term that improves sampling speed. This, however, is already discovered in prior work cited in the paper, which reduces the technical novelty. The experimental section is good, and after the rebuttal, complete, but the mechanism of the MAP estimate (which is used in classical models) doesn't seem to be the one here. Better understanding of why GAN regularization works would help the paper greatly.

**Justification For Why Not Lower Score:**

N/A

**Metareview: Summary, Strengths And Weaknesses:**

The authors propose restoration-based generative models, which comprise two components: 1) using a prior term based on GANs, and 2) . The paper was greatly improved during the rebuttal process, with better presentation, more experiments, and correction of mistakes. The main motivation of the paper -- that the new objective is based on a MAP estimate -- is tenuous. Specifically, the GAN "prior" reposes on the assumption that the discriminator can provide an accurate estimate of the Jensen-Shannon (JS) divergence, which is not true for discriminators not trained until convergence. A more accurate presentation would be that GAN regularization greatly decreases the number of sampling steps